# A BET family protein degrader provokes senolysis by targeting NHEJ and autophagy in senescent cells

Masahiro Wakita[1], Akiko Takahashi[2], Osamu Sano[3], Tze Mun Loo[2], Yoshinori Imai[2], Megumi Narukawa[1], Hidehisa Iwata [3], Tatsuyuki Matsudaira[1,4], Shimpei Kawamoto[1], Naoko Ohtani[5], Tamotsu Yoshimori[6] & Eiji Hara [1,2,4✉]

Although cellular senescence acts primarily as a tumour suppression mechanism, the accumulation of senescent cells *in vivo* eventually exerts deleterious side effects through inflammatory/tumour-promoting factor secretion. Thus, the development of new drugs that cause the specific elimination of senescent cells, termed senolysis, is anticipated. Here, by an unbiased high-throughput screening of chemical compounds and a bio-functional analysis, we identify BET family protein degrader (BETd) as a promising senolytic drug. BETd provokes senolysis through two independent but integrated pathways; the attenuation of non-homologous end joining (NHEJ), and the up-regulation of autophagic gene expression. BETd treatment eliminates senescent hepatic stellate cells in obese mouse livers, accompanied by the reduction of liver cancer development. Furthermore, the elimination of chemotherapy-induced senescent cells by BETd increases the efficacy of chemotherapy against xenograft tumours in immunocompromised mice. These results reveal the vulnerability of senescent cells and open up possibilities for its control.

[1] Research Institute for Microbial Diseases (RIMD), Osaka University, Suita 565-0871, Japan. [2] Cancer Institute, Japanese Foundation for Cancer Research, Tokyo 135-8550, Japan. [3] BioMolecular Research Laboratories, Takeda Pharmaceutical Company Ltd., Fujisawa 251-8555, Japan. [4] Immunology Frontier Research Center (IFReC), Osaka University, Suita 565-0871, Japan. [5] Graduate School of Medicine, Osaka City University, Osaka 545-8585, Japan. [6] Graduate School of Medicine, Osaka University, Suita 565-0871, Japan. ✉email: ehara@biken.osaka-u.ac.jp

Oncogenic proliferative signals are coupled to a variety of growth inhibitory processes, such as the induction of apoptotic cell death and stable cell-cycle arrest, in phenomena termed cellular senescence[1–3]. Both apoptosis and cellular senescence are considered to serve as important safeguards against neoplasia[2–4]. However, unlike apoptotic cells, senescent cells are viable for long periods of time and thereby accumulate with age in various organs and tissues[5–7]. Moreover, it has recently become apparent that senescent cells are not merely nondividing, but eventually develop a secretory profile composed of pro-inflammatory cytokines, chemokines, and extracellular matrix-degrading proteases, a typical signature termed the senescence-associated secretory phenotype (SASP)[8–11]. Although SASP reportedly plays some beneficial roles, it also exhibits deleterious side effects such as chronic-inflammation and/or tumourigenesis, depending on the biological context[2,3,11–13]. Thus, although cellular senescence primarily acts as a tumour suppression mechanism, the accumulation of senescent cells in aged tissues may eventually promote the age-related decline of organ function and/or associated diseases, such as cancer[2,3]. Indeed, the clearance of p16[INK4a]-positive senescent cells from aged transgenic mice reportedly delays the onset of various age-related dysfunctions, such as sarcopenia, cataracts, atherosclerosis, loss of adipose tissue, and tumourigenesis, thus extending the healthy lifespan[12]. Along similar lines, the elimination of therapy-induced senescent cells reduced several side-effects of chemotherapy and even cancer recurrence in mice[14]. Thus, it is anticipated that the removal of senescent cells could prevent the toxicity of anticancer treatments and enhance the therapeutic benefits[15,16].

Senolytic drugs, which specifically induce cell death in senescent cells, are likely to represent a new therapeutic avenue, and several candidate drugs were identified using a bioinformatics approach[15]. However, the senolytic drugs identified to date were not discovered by a truly unbiased high-throughput screening (HTS) method, and thus appear to have limitations in clinical applications[15,17–19]. For example, in a phase II study of ABT263 (a specific inhibitor of the anti-apoptotic proteins BCL2 and BCL-xL)[19] for the treatment of advanced and recurrent small-cell lung carcinoma patients, transient thrombocytopenia and neutropenia were reported as side-effects[15]. Furthermore, the combination of dasatinib and quercetin (D + Q), another previously reported senolytic drug[20], apparently exhibited remarkable cell-type specificity[15], although its mechanisms of action remain obscure. Therefore, the identification of more effective senolytic drugs and the elucidation of their mechanisms of action are required, towards a better strategy for the removal of senescent cells in vivo[15,16].

In this study, we identify bromodomain and extra-terminal domain (BET) family protein inhibitor (BETi)[21,22] as a promising senolytic drug. The blockade of BRD4, a BET family protein, by chemical inhibitors or RNA interference robustly provokes senolysis. This is due, at least in part, to the combined effects of the attenuation of non-homologous end joining (NHEJ) repair and the activation of autophagic pathway in senescent cells. These results reveal the cellular vulnerability of senescent cells, and provide valuable insights into the resistance of senescent cells to death and possibilities for its control.

## Results

**High-throughput screening of senolytic drugs**. To identify more effective senolytic drugs, we conducted an unbiased HTS of a chemical compound library consisting of around 47,000 small molecules, in an open innovation drug discovery program for academic researchers sponsored by the Takeda Pharmaceutical Company (RINGO-T project). Early passage human diploid fibroblasts (HDFs) expressing a tamoxifen-regulated form of oncogenic Ras were treated with or without 4-hydroxytamoxifen (4-OHT) for 8 days, and then incubated with each chemical compound of the library for 72 h (Fig. 1a). A cell viability analysis based on measurements of the ATP levels and Caspase3/7 activity revealed that 15 small molecules preferentially induce cell death in Ras-induced senescent cells, as compared to the control non-senescent cells. Intriguingly, although we were unable to find previously reported senolytic drugs in our HTS, four top ranking hit molecules (JQ1, I-BET151, I-BET762, and PFI-1) turned out to be inhibitors of bromodomain and extra-terminal (BET) family proteins[21,23], and are currently in advanced clinical development as a new class of therapeutics for the treatment of human cancers[23].

To further confirm and extend our HTS results, the senolytic activities of JQ1[23] (the most frequently used BET inhibitor (BETi) in published studies), OTX015[23] (a more potent analogue of JQ1), and ARV825[24] (a recently developed small-molecule BET degrader (BETd)) were compared to those of previously reported senolytic drugs, such as ABT263[19], 17-DMAG[25] (HSP90 inhibitor), and the combination of dasatinib and quercetin (D + Q)[20]. Notably, although all of the tested drugs certainly showed some senolytic activity, ARV825 had the strongest senolytic activity and killed senescent HDFs at a five to ten nanomolar concentration, regardless of the manner of cellular senescence induction (Fig. 1b to e, and Supplementary Fig. 1). This is consistent with the observation that the levels of the BRD3 and BRD4 BET family proteins were significantly reduced by the 10 nM ARV825 concentration treatment in senescent cells (Fig. 1c and e). Similar effects of ARV825 were also observed across cell types and species (Supplementary Fig. 2). Notably, although a long term treatment with ABT263 caused apoptotic cell death in confluence (contact-inhibition)-induced reversible cell-cycle arrest (quiescence), even at an optimal concentration for senolysis (500 nM), this was not the case with the ARV825 treatment (Supplementary Fig. 3), further illustrating the specificity of ARV825 towards senescent cells. Therefore, we used 10 nM ARV825 for all subsequent experiments in the present study. Note that although the Senescence-Associated β-galactosidase (SA β-gal) activity is a widely used senescence marker, it is also induced when HDFs are rendered quiescent by contact-inhibition or serum starvation[26,27]. Therefore, we decided to use a combination of other senescence markers, such as the induction of p21[Waf1/Cip1/Sdi1] expression[28], the reduction of Lamin B1 expression[29,30], and the increase of DNA damage response (DDR) marker, such as 53BP1 foci formation[27], throughout this study.

**ARV825 provokes senolysis in vivo**. To determine whether ARV825 exhibits senolytic activity in vivo and alleviates the deleterious side effects of senescent cells in vivo, we used a previously developed obesity-induced hepatocellular carcinoma (HCC) mouse model[13]. In these mice, the increased level of deoxycholic acid (DCA), a gut bacterial metabolite known to cause DNA damage, provokes cellular senescence and SASP in hepatic stellate cells (HSCs), which in turn promote HCC development in neighbouring hepatocytes through SASP[13]. Indeed, the ARV825 treatment substantially reduced HCC development (Fig. 2a, b), coincident with the reduction of the senescent HSC number in the obesity-induced HCC mouse model, as judged by the immunofluorescence (IF) staining analysis using the combination of antibodies against αSMA (activated HSCs marker) and p21 (senescence marker) or IL6/Groα (SASP markers)[13] (Fig. 2c). These results, along with the data from cultured senescent HSCs provoked by DCA treatment

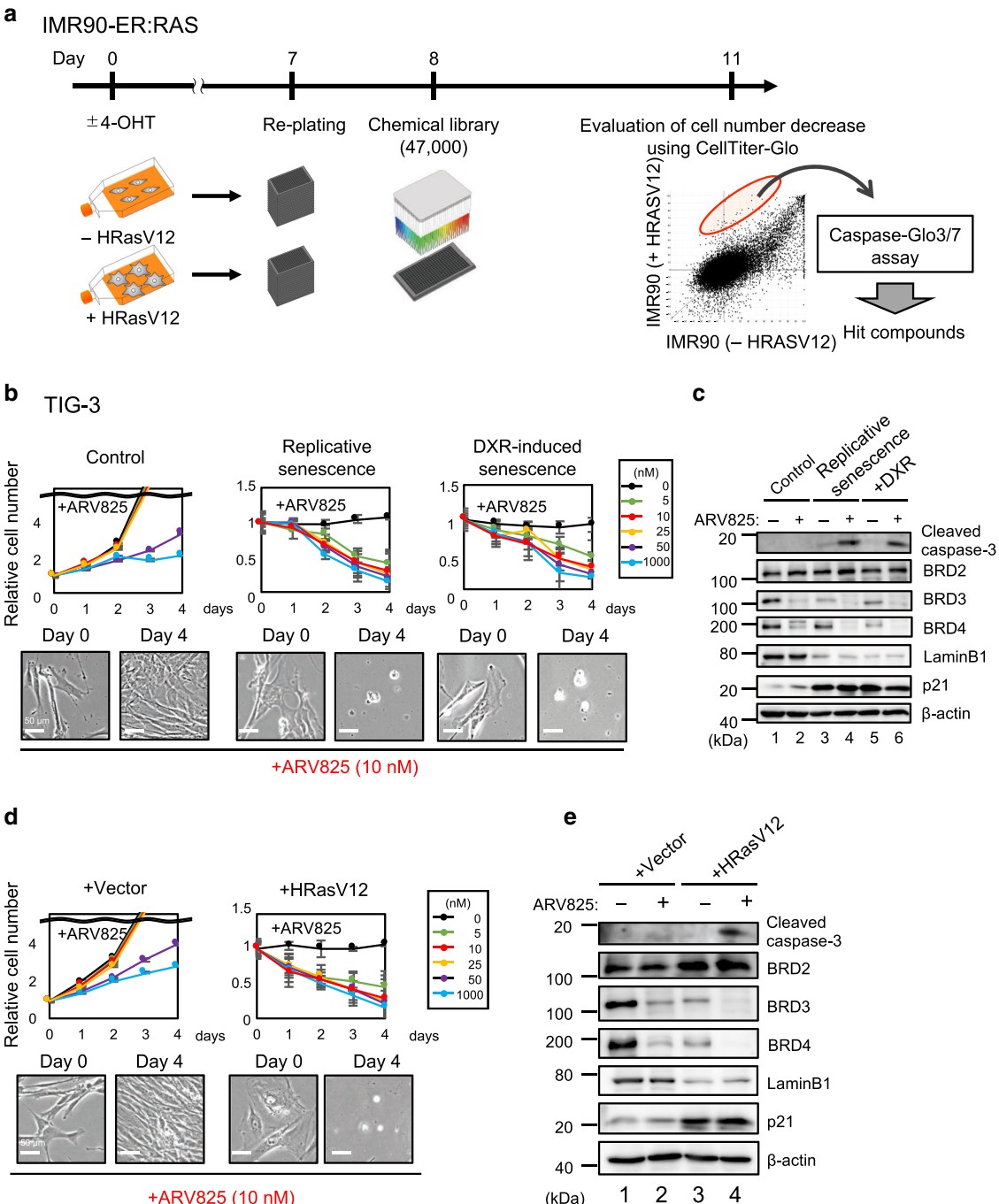

**Fig. 1 HTS identified BET inhibitors as potent senolytic compounds. a** Outline of the high-throughput screening (HTS) procedure for identifying senolytic compound. IMR90-ER:Ras cells[44] were stimulated with or without 4OHT for 7 days to induce senescence. These cells were plated on 384-well plates and incubated for another 24 hrs, followed by treatment with Takeda chemical compound library consisting of 47, 000 small molecules for 3 days. These cells were then subjected to cell viability analysis using CellTiter-Glo Luminescent Cell Viability assay system (Promega), followed by apoptosis analysis using Caspase-Glo3/7 Assay system (Promega). **b–e** Early passage pre-senescent (control) normal human diploid fibroblasts (TIG-3 cells) were rendered senescent by serial passage (replicative senescence), treatment with 250 ng/ml doxorubicin for 10 days (+DXR) (**b**) or infection with retrovirus encoding oncogenic Ras (+HRasV12) (**d**). These senescent cells and control pre-senescent cells (control (**b**) and +Vector (**d**)) were then incubated with increasing concentration of ARV825 indicated at right for 4 days. Relative cell number was counted throughout the experiments and representative photographs of the cells in the indicated culture conditions are shown at the bottom of the **b**, **d**. For all graphs, error bars indicate mean ± s.d. (n = 3) (**b**, **d**). Cells treated with or without 10 nM ARV825 for 4 days were subjected to western blotting using the antibodies shown on the right (**c**, **e**). β-actin was used as a loading control. The representative data from three independent experiments were shown (**b–e**). Source data are provided as a Source Data file.

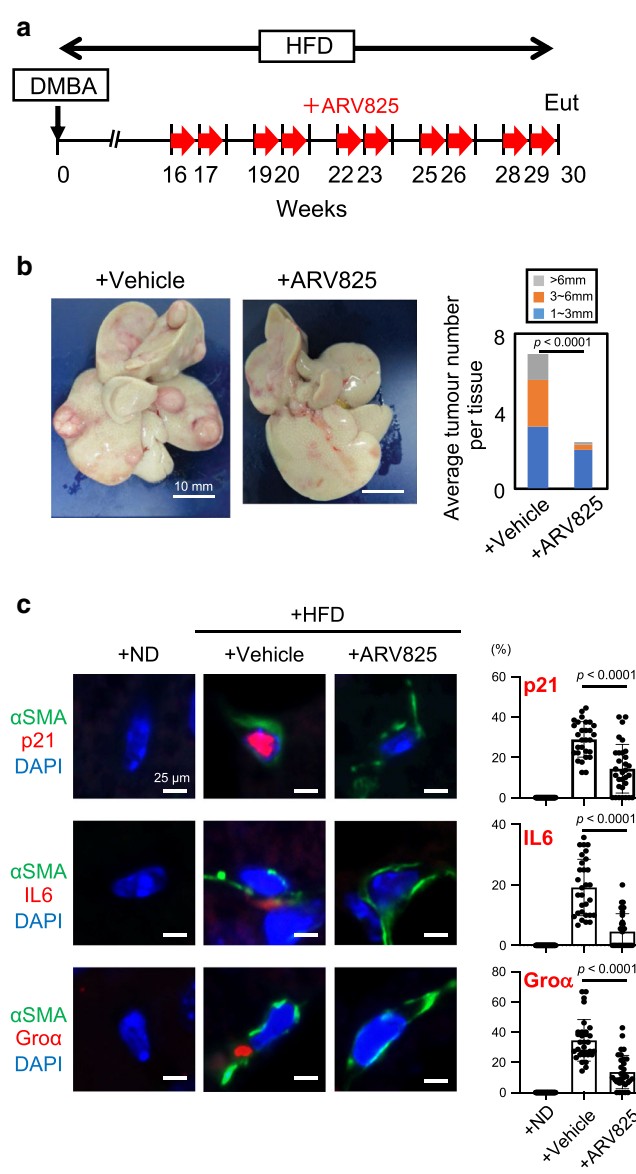

**Fig. 2 BETd reduces obesity-associated HCC development in mice on deleterious side effects of senescent cells in vivo. a** Timeline of the experimental procedure[13]. Eut, euthanasia. Red arrow indicates a daily treatment with ARV825 (injected intraperitoneally with 5 mg/kg) or vehicle for 5 consecutive days. **b** Representative macroscopic photographs of livers. The average liver tumor numbers and their relative size distribution were shown right (HFD + vehicle, $n = 20$; HFD + ARV825, $n = 16$). **c** Immunofluorescence of liver section. HSCs were visualized by αSMA staining (green) and DNA was stained by 4′ 6,-diamidino-2-phenylindole (DAPI; blue)[13]. The histograms indicate the percentages of αSMA-expressing cells that were positive for indicated markers. The number of cells scored per group was as follows: $n = 329$ (p21 + ND), $n = 355$ (p21 + Vehicle), $n = 351$ (p21 + ARV825), $n = 309$ (Groα + ND), $n = 286$ (+Vehicle), $n = 385$ (+ARV825), $n = 320$ (IL-6), $n = 460$ (+Vehicle), $n = 337$ (+ARV825) over three biological independent animals. Error bars indicate mean ± standard deviation (s.d.). Statistical significance was determined with two-tailed unpaired Student's t-test (**b**), one-way ANOVA followed by Holm-Sidak multiple comparison test (**c**), P values < 0.05 were considered significant. Source data are provided as a Source Data file.

(Supplementary Fig. 2e, f), strongly suggest that ARV825 has the potential to eliminate senescent HSCs in vivo, thereby reducing HCC development, at least in our obesity mouse model. To further explore the beneficial impact of ARV825 on the side effects of senescent cells in vivo, we next tested the senolytic activity of ARV825 in therapy-induced senescent cells[14]. Notably, cultured senescent HCT116 cells (a human colon cancer cell line) provoked by doxorubicin (DXR) treatment were preferentially killed by the ARV825 treatment, as compared to control HCT116 cells (Fig. 3a, b). Furthermore, the tumour size was significantly reduced when HCT116 xenograft mice were co-treated with DXR and ARV825 as compared to DXR treatment alone, coincident with a significant reduction of p21$^{Waf1/Cip1/Sdi1}$ expression and 53BP1 foci formation, senescence markers (Fig. 3c–e). Collectively, these results strongly suggest that ARV825 also has senolytic activity in vivo and could have some beneficial impact on the deleterious side effects of senescent cells, depending on the biological context.

**ARV825 down-regulates *xrcc4* expression in senescent cells**. To further consolidate this idea, we next explored how ARV825 preferentially kills senescent cells. ARV825 is a heterobifunctional PROTAC (Proteolysis Targeting Chimera) that recruits BET family proteins to the E3 ubiquitin ligase CERE-BLON, leading to the fast, efficient, and prolonged degradation of BET family proteins[24]. Although HDFs express three BET family proteins, BRD2, BRD3, and BRD4, the ARV825 treatment reduced the levels of BRD3 and BRD4, but not BRD2, in senescent HDFs (Fig. 1c, e). These results, in conjunction with the observation that the siRNA-based depletion of BRD4, but neither BRD2 nor BRD3, robustly provoked senolysis in HDFs (Supplementary Fig. 4), indicate that BRD4 is the major senolysis target of ARV825, at least in HDFs. Note that TIG-3 cells express both the long and short isoforms of BRD4[22], and both ARV825 and the above-mentioned siRNA against BRD4 targeted both isoforms of BRD4 in senescent cells (Supplementary Fig. 5a). Thus, we next asked which isoform is more responsible for protecting senescent cells from senolysis. Intriguingly, the knockdown of the long isoform, but not the short isoform, efficiently provoked senolysis, indicating that the long isoform containing the carboxy-terminal domain (CTD) plays more important roles in protecting senescent cells from senolysis (Supplementary Fig. 5b, c).

Since BRD4 reportedly resides at and upregulates super-enhancer regions[21], which are often upstream of oncogenes such as *c-myc*, *bcl-xl*, and *bcl-6*, we performed an RNA sequencing (RNA-seq) analysis to identify the oncogenes with expression that is downregulated by the ARV825 treatment in senescent HDFs. Unexpectedly, we were unable to detect any substantial declines in the expression of previously reported BRD4-targeted oncogenes (Fig. 4a), perhaps because the expression of many, if not most, oncogenes is silenced by multiple mechanisms, such as the formation of senescence-associated heterochromatic foci (SAHF)[2], in senescent cells. Instead, we found that the expression levels of the *xrcc4* gene substantially declined upon the treatment of senescent HDFs with ARV825 (Fig. 4a and Supplementary Fig. 6).

**ARV825 inhibits the NHEJ repair machinery in senescent cells.** The protein encoded by the *xrcc4* gene (XRCC4) forms a complex with DNA ligase IV (LIG4) and plays an important role in non-homologous end joining (NHEJ) repair for DNA double-strand breaks (DSBs)[31]. Since NHEJ, but not homologous recombination (HR), is the major DNA repair mechanism for DSBs in non-dividing cells[31], such as senescent cells, we wondered if ARV825

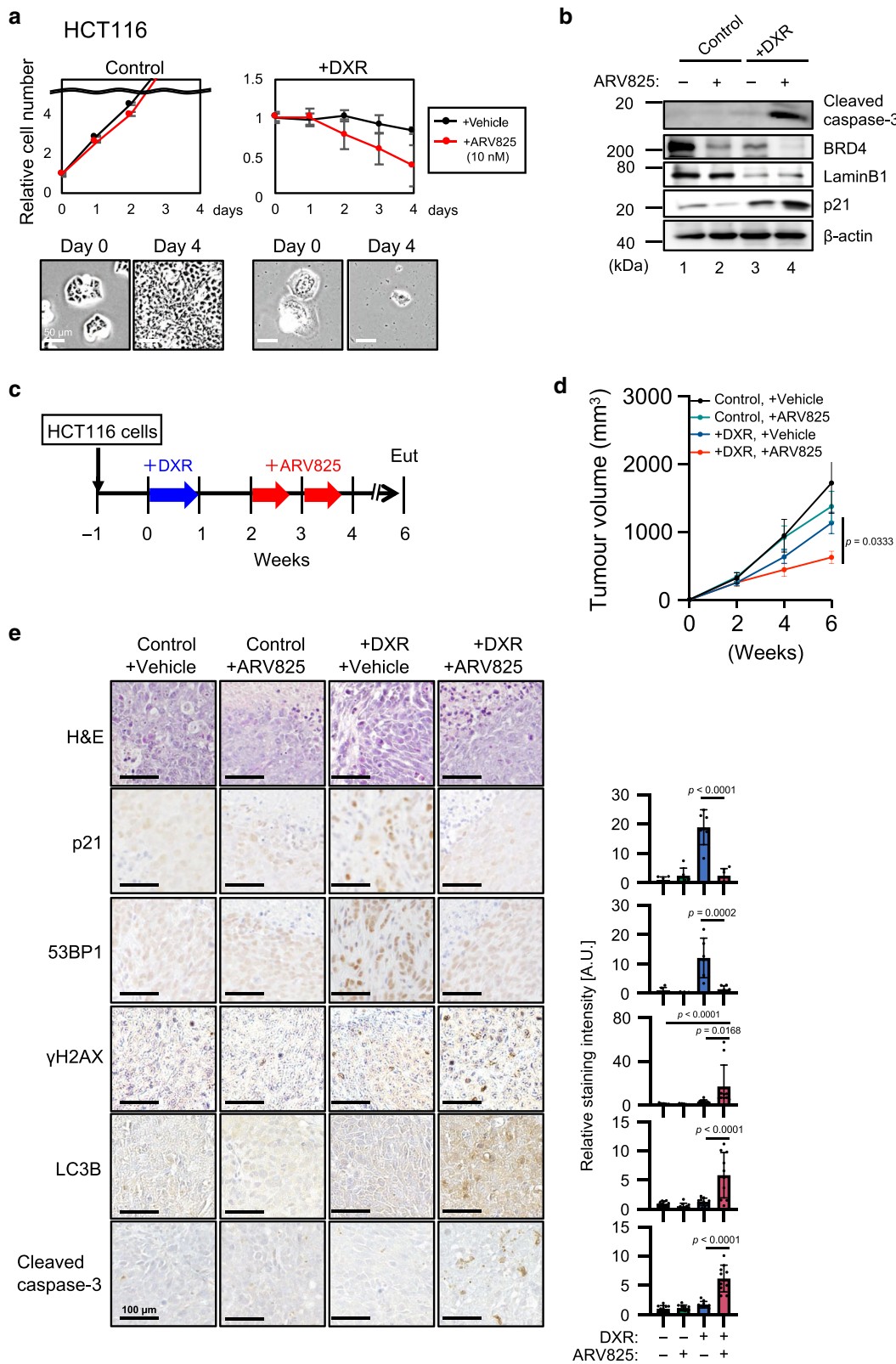

causes senolysis by exacerbating DSBs in senescent cells. Indeed, the ARV825 treatment caused the elevation of DSBs in senescent HDFs, as judged by the γH2AX foci formation assay and the neutral comet assay, regardless of how cellular senescence was induced (Fig. 4b and c, Supplementary Fig. 7). These effects were phenocopied by the siRNA-based depletion of XRCC4 or BRD4 in senescent HDFs (Supplementary Figs. 8 and 9), implying that

ARV825 may cause senolysis by inhibiting the NHEJ machinery through blocking *xrcc4* gene expression in senescent cells. Unexpectedly, however, XRCC4 overexpression failed to rescue the ARV825-induced DSBs and senolysis (Supplementary Fig. 10a to d), raising the possibility that other mechanisms are also involved in the inhibitory action of ARV825 on the NHEJ machinery.

**Fig. 3 BETd increases the efficacy of chemotherapy against xenograft tumours in mice. a, b** Control and senescent HCT116 induced by treatment with 200 ng/ml doxorubicin for 10 days (+DXR) were incubated with 10 nM ARV825 or vehicle for 4 days. Relative cell number was counted throughout the experiments and representative photographs of the cells in the indicated culture conditions are shown at the bottom of the **a**. Error bars indicate mean ± s.d. ($n = 3$) (**a**). Cells treated with 10 nM ARV825 (+) or vehicle (−) for 4 days were subjected to western blotting using the antibodies shown on the right (**b**). Representative data from three independent experiments were shown (**a, b**). **c–e** Timeline of the experimental procedure. Blue arrow indicates the daily treatment with DXR (injected intraperitoneally with 1 mg/kg) or vehicle for 7 consecutive days. Red arrow indicates a daily treatment with ARV825 (injected intraperitoneally with 5 mg/kg) or vehicle for 5 consecutive days (**c**). Tumor volume was calculated weekly from $n = 4$ (control + Vehicle), $n = 5$ (control + ARV825), $n = 5$ (+DXR + Vehicle), $n = 4$ (+DXR + ARV825) biologically independent animals using calipers (**d**). Error bars indicate mean± s.e.m (**d**). Serial sections of the biopsy samples of xenografted tumours were subjected to hematoxylin and eosin (HE) staining, immunohistochemistry for senescence markers (p21$^{Waf1/Cip1/Sdi1}$ and 53BP1), DNA damage marker (γH2AX), autophagy marker (LC3B) and apoptosis marker (Cleaved caspase-3) (**e**). The histogram shown at the right of the panel indicates the percentages of positively stained areas (**e**). The number of positive signals in the images was calculated from $n = 6$ (p21 and 53BP1) or $n = 10$ (γH2AX, LC3B, and Cleaved caspase-3) biologically independent tissue sections over four biological independent animals. Error bars indicate mean ± s.d. (**e**). Statistical significance was determined with a Kruskal–Wallis followed by Dunn's multiple comparison test (**d**), or one-way ANOVA followed by Tukey multiple comparison test (**e**). P values < 0.05 were considered significant. Source data are provided as a Source Data file.

Indeed, emerging evidence has indicated that BRD4 has the potential to bind 53BP1, a DNA damage response protein serving as an adaptor for the assembly and activation of the DNA repair machinery, and recruit it to DSB sites, thereby facilitating NHEJ in a transcriptionally independent manner[32,33]. In concordance with this notion, the interaction between BRD4 and 53BP1 was observed in senescent cells, but was reduced when senescent HDFs were treated with ARV825 (Fig. 5a). Moreover, 53BP1 foci formation was abolished upon ARV825 treatment or BRD4 depletion, accompanied by the exacerbation of DSBs in senescent HDFs (Fig. 4b and c, Supplementary Figs. 7 and 9). However, consistent with the fact that there are no reports showing that XRCC4 is required for the recruitment of 53BP1 to DSB sites, the siRNA-based depletion of XRCC4 exacerbated the formation of DSBs without blocking the 53BP1 localisation to DSB sites in senescent cells (Supplementary Fig. 8d and e). Conversely, the overexpression of XRCC4 failed to rescue 53BP1 foci formation in senescent cells treated with ARV825 (Supplementary Fig. 10). Moreover, the effects of ARV825 were phenocopied by the depletion of 53BP1 in senescent HDFs (Fig. 5b-f). Thus, considering the published data[32,33] and our findings, it is very likely that ARV825 provokes senolysis by exacerbating DSBs through targeting at least two independent mechanisms of the NHEJ machinery in senescent cells: (1) inhibiting the *xrcc4* gene expression and (2) blocking the recruitment of 53BP1 to DSB sites.

**Autophagy is required for ARV825-induced senolysis.** To substantiate the idea that ARV825 provokes senolysis through blocking the NHEJ machinery, we next tested if the senolytic effect of the ARV825 treatment could be phenocopied by a treatment with NU7026, a potent DNA-PK inhibitor known to block NHEJ[31]. Unexpectedly, the NU7026 treatment failed to induce senolysis in senescent HDFs, although the DSBs were increased to the same levels as those seen with the ARV825 treatment (Fig. 6a, b). These results suggest that mechanisms other than DSB formation are also involved in the ARV825-induced senolysis, and raise the question of how the siRNA-mediated depletion of XRCC4 or 53BP1 caused senolysis (Fig. 5d and Supplementary Fig. 8c).

The cationic lipids used for the siRNA-mediated depletion are known to induce autophagy[34], and BRD4 inhibition has recently been reported to induce autophagic gene expression[22]. Therefore, we tested if the activation of the autophagy machinery[35] is required for the execution of the ARV825-induced senolysis. Indeed, consistent with a previous report[22], the expression of a series of autophagic genes was upregulated by the ARV825 treatment in senescent HDFs (Fig. 4a, Supplementary Fig. 11).

Furthermore, the number of LC3 puncta, a sign of autophagy activation[36], was substantially increased by the ARV825 treatment or by the siRNA-mediated depletion of XRCC4 or 53BP1 in senescent HDFs (Fig. 6c, Supplementary Fig. 12), and the co-treatment with NU7026 and cationic lipids provoked increases of LC3 puncta and senolysis in senescent HDFs (Fig. 6a, c). Importantly, this was not the case when senescent HDFs were separately treated with either the cationic lipids or the DNA-PK inhibitor (Fig. 6a, c), indicating that the treatment with cationic lipids and the exacerbation of DSBs are both required for the full activation of the autophagic machinery in senescent cells. This is consistent with accumulating evidence that DSBs activate autophagy[37]. Conversely, the treatment of senescent HDFs with autophagy inhibitors, such as bafilomycin A1 or chloroquine, substantially reduced the ARV825-induced senolysis (Fig. 7a, Supplementary Fig. 13a).

Notably, the tumour suppressive effect of ARV825 on HCT116 xenograft mice treated with DXR coincided with an increase of cleaved caspase-3 (an apoptosis marker), together with γH2AX (a DNA damage marker), and LC3B (an autophagy marker) in tumour tissues (Fig. 3e). Furthermore, the combined use of chloroquine with ARV825 significantly attenuated the tumour suppressive effect of ARV825 in HCT116 xenograft mice treated with DXR (Supplementary Fig. 13b). Together, these results strongly suggest that BRD4 inhibition provokes senolysis by activating the autophagy machinery, through the exacerbation of DSBs and the up-regulated expression of several autophagic genes in senescent cells.

**Autophagy machinery preceded the apoptosis in senolysis.** We next explored how autophagy induces senolysis. It should be noted that the levels of the cleaved form of caspase 3, a sign of apoptosis, correlated well with those of senolysis (Fig. 1c, e, and Supplementary Fig. 2b, d, and f), and a treatment with a caspase inhibitor (Q-VD-OHT) attenuated the senolysis provoked by ARV825, without reducing the number of LC3 puncta in senescent HDFs (Fig. 7a and b). On the other hand, although the bafilomycin A1 treatment attenuated the ARV825-induced senolysis, this was accompanied by a substantial reduction of the activated caspase 3 levels (Fig. 7a and c). Collectively, these results indicate that the autophagy machinery[35] preceded the apoptosis in ARV825-induced senolysis. Moreover, the co-treatment with an autophagy inhibitor and a caspase inhibitor attenuated the senolysis at similar levels, as compared with the single treatment with each inhibitor, in ARV825-treated senescent cells (Fig. 7a), further supporting the above idea that autophagy and apoptosis act on the same pathway in ARV825-induced senolysis. These results, in conjunction with the recent

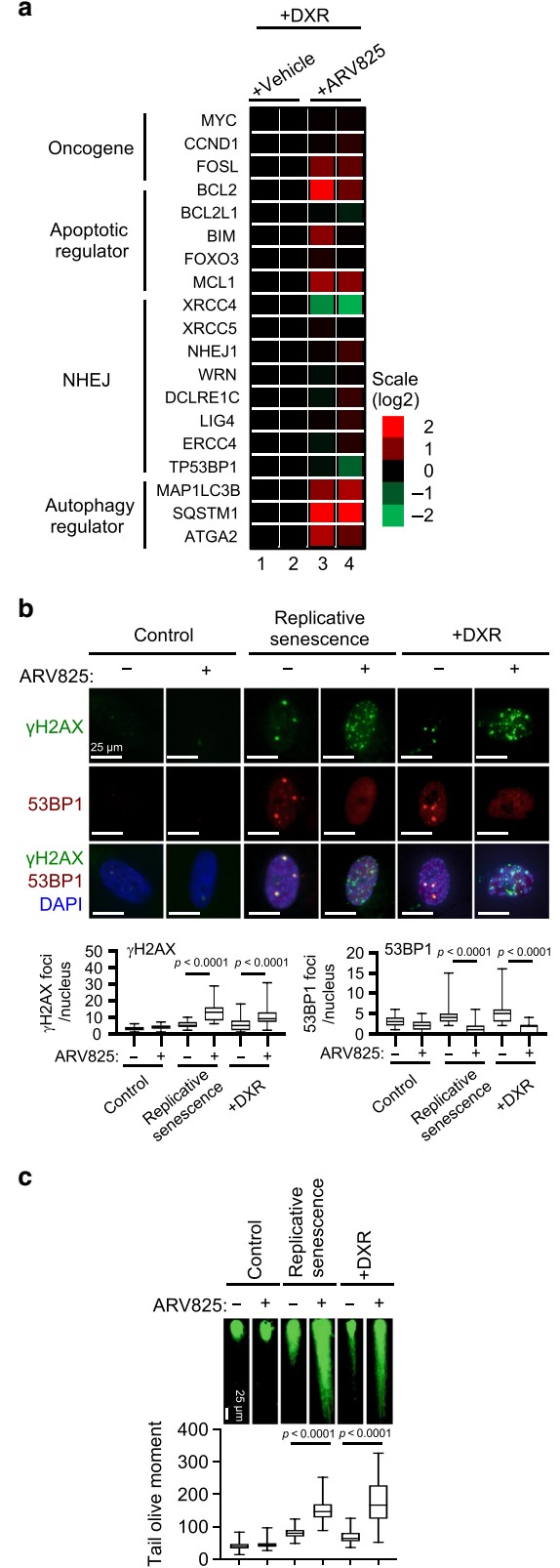

**Fig. 4 BETd accelerates DSBs in senescent cells. a** Heat map representation of the expression of genes (from RNA-seq experiments) upon treatment of DXR-induced senescent TIG-3 cells with 10 nM ARV825 or vehicle for 2 days. The heat map key indicates log2-fold changes from baseline. **b**, **c** Early passage (control) TIG-3 cells were rendered senescent by serial passage (replicative senescence) or treatment with 250 ng/ml doxorubicin for 10 days (+DXR). These cells and control cells were treated with 10 nM ARV825 (+) or vehicle (−) for 4 days and were then subjected to immunofluorescence staining using the antibodies shown on the left (**b**) or to neutral comet assay (**c**). The number of γH2AX or 53BP1 foci, above threshold intensity per nucleus ($n = 50$), was quantified and shown at the bottom of the **b**. The average tail olive moments were shown at the bottom of the **c**. Each box indicates the median and 25% and 75% quantiles, and the whiskers represent the minimum and maximum observations (**b**, **c**). Error bars indicate mean ± s.d. The representative data from three independent experiments were shown (**b**, **c**). Statistical significance was determined with one-way ANOVA followed by Holm-Sidak multiple comparison test (**b**, **c**). P values < 0.05 were considered significant. Source data are provided as a Source Data file.

## Discussion

In the present study, we have identified BETd as a promising senolytic drug, through an unbiased high-throughput screening of chemical compound libraries consisting of around 47,000 small molecules. The blockade of BRD4 by ARV825, a recently developed small-molecule BETd[24], robustly provokes autophagy-induced apoptosis in senescent cells. This is due, at least in part, to two independent but integrated pathways: (1) exacerbation of DNA double-strand breaks (DSBs) by blocking non-homologous end joining (NHEJ) repair, and (2) up-regulation of autophagic gene expression (see Model in Fig. 7d). It should be noted, however, that all aspects of the BRD4 inhibition-induced senolysis might not be explained by the factors described here. In this respect, it is noteworthy that BETi reportedly downregulates SASP factor expression in senescent cells[39] and several SASP factors, such as GM-CSF[8] or PDGF[40], are known to promote cell survival. Thus, it is tempting to speculate that the downregulation of SASP factor expression may also contribute to the BETi-induced senolysis, depending on the biological context.

Since our current knowledge of senescent cells is largely based on cell culture systems, and efforts to characterise senescent cells in vivo have been limited due to the lack of absolute markers for senescent cells[1], it remains unclear whether the senolytic effects of BETd are common to all types of senescent cells in vivo. Moreover, unlike proliferating cells, healthy non-dividing cells such as quiescent cells also rely on NHEJ for DSB repair[31]. Thus, the therapeutic applications of BETd should be approached with caution. However, since DSBs occur less frequently in quiescent cells, the blockade of NHEJ is likely to have a relatively minor impact in quiescent cells, thus explaining why the ARV825 treatment provokes cell death preferentially in senescent cells, rather than quiescent cells (Fig. 1b, d and Supplementary Fig. 3). Furthermore, OTX015, a BETi, reportedly has a favorable safety profile, with clinical activity observed in NUT midline carcinoma[41]. Therefore, the development of more potent BETd molecules is expected to produce effective senolytic drugs, at least in certain settings.

Unlike previously reported senolytic drugs[17,19,20,25,42,43] (Fig. 1, Supplementary Fig. 1), ARV825 exhibits robust senolysis activity even at nanomolar concentrations (5–10 nM) (Fig. 1). These results, in conjunction with the observation that the

observation that the aberrant activation of autophagy results in caspase-dependent apoptotic cell death[38], strongly suggest that BRD4 inhibition provokes autophagy-induced apoptosis by exacerbating DSBs and upregulating the expression of several autophagic genes in senescent cells, thereby causing senolysis (Model in Fig. 7d).

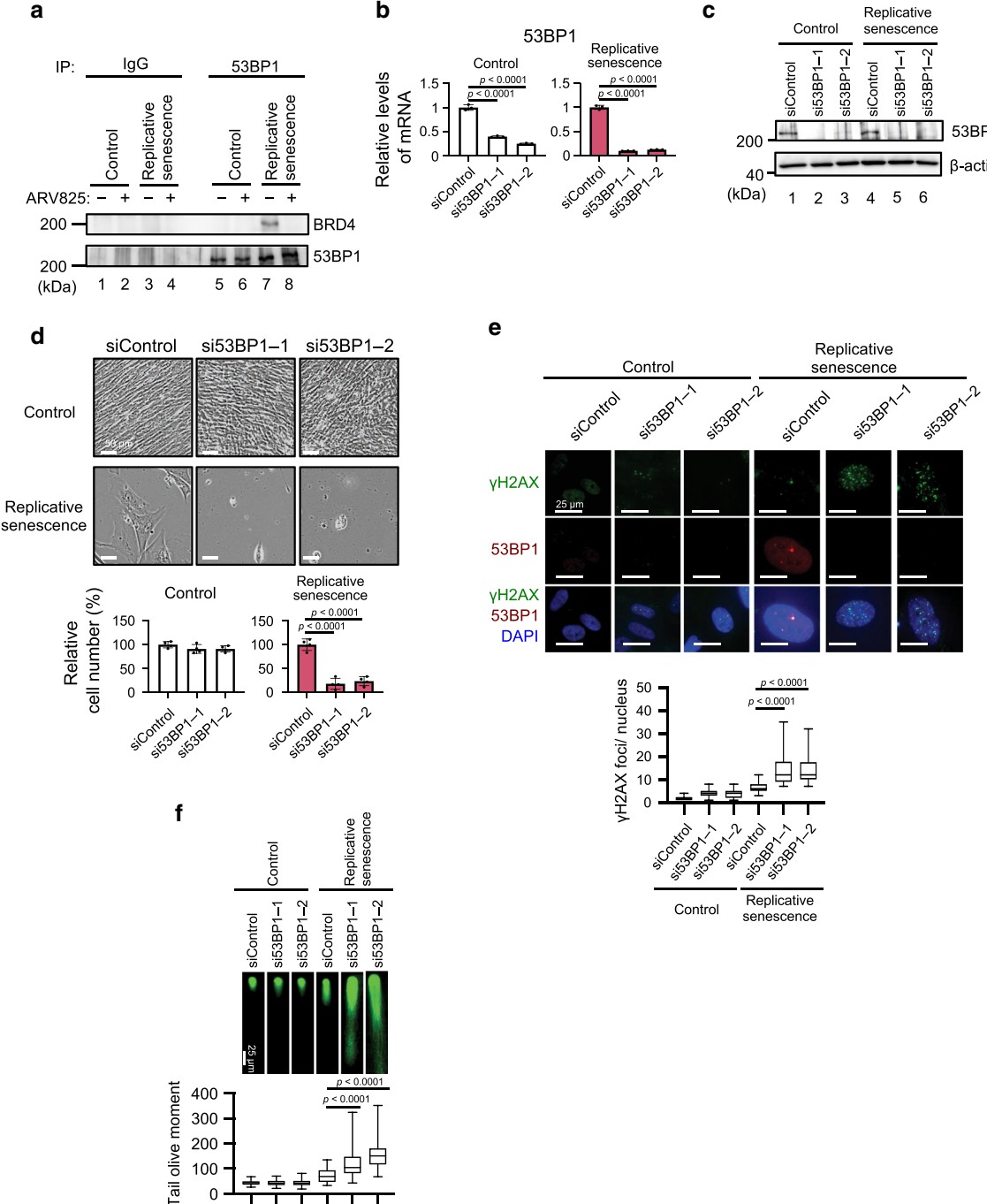

**Fig. 5 BETd blocks 53BP1 function in senescent cells. a** Early passage (control) or late passage (replicative senescence) TIG-3 cells were treated with 10 nM ARV825 (+) or vehicle (−) for 4 days. These cells were then subjected to western blotting using the antibodies shown on the right after immunoprecipitation with the antibodies shown at the top of the panel (IP:). **b–f** Early passage (control) or late passage (replicative senescence) TIG-3 cells were transfected with previously validated two different siRNA oligos against 53BP1 or control siRNA oligo twice at 2 days intervals. These cells were then subjected to RT-qPCR analysis (**b**), western blotting analysis (**c**), the cell proliferation analysis (**d**), immunofluorescence staining using the antibodies shown on the left (**e**) or neutral comet assay (**f**). Representative photographs of the cells in the indicated culture conditions are shown and the histogram shown at the bottom of the panel indicates the relative cell number (**d**). The number of γH2AX foci, above threshold intensity per nucleus (n = 50) was quantified and shown at the bottom of the **e**. The average tail olive moments were shown at the bottom of the **f**. Each box indicates the median and 25% and 75% quantiles, and the whiskers represent the minimum and maximum observations (**e**, **f**). For all graphs, error bars indicate mean ± s.d. (n = 3 for **b**, n = 4 for **d**) and the representative data from three independent experiments were shown. Statistical significance was determined with two-tailed unpaired Student's *t*-test (**b**), one-way ANOVA followed by Tukey multiple comparison test (**d**), one-way ANOVA followed by Holm-Sidak multiple comparison test (**e**, **f**). *P* values < 0.05 were considered significant. Source data are provided as a Source Data file.

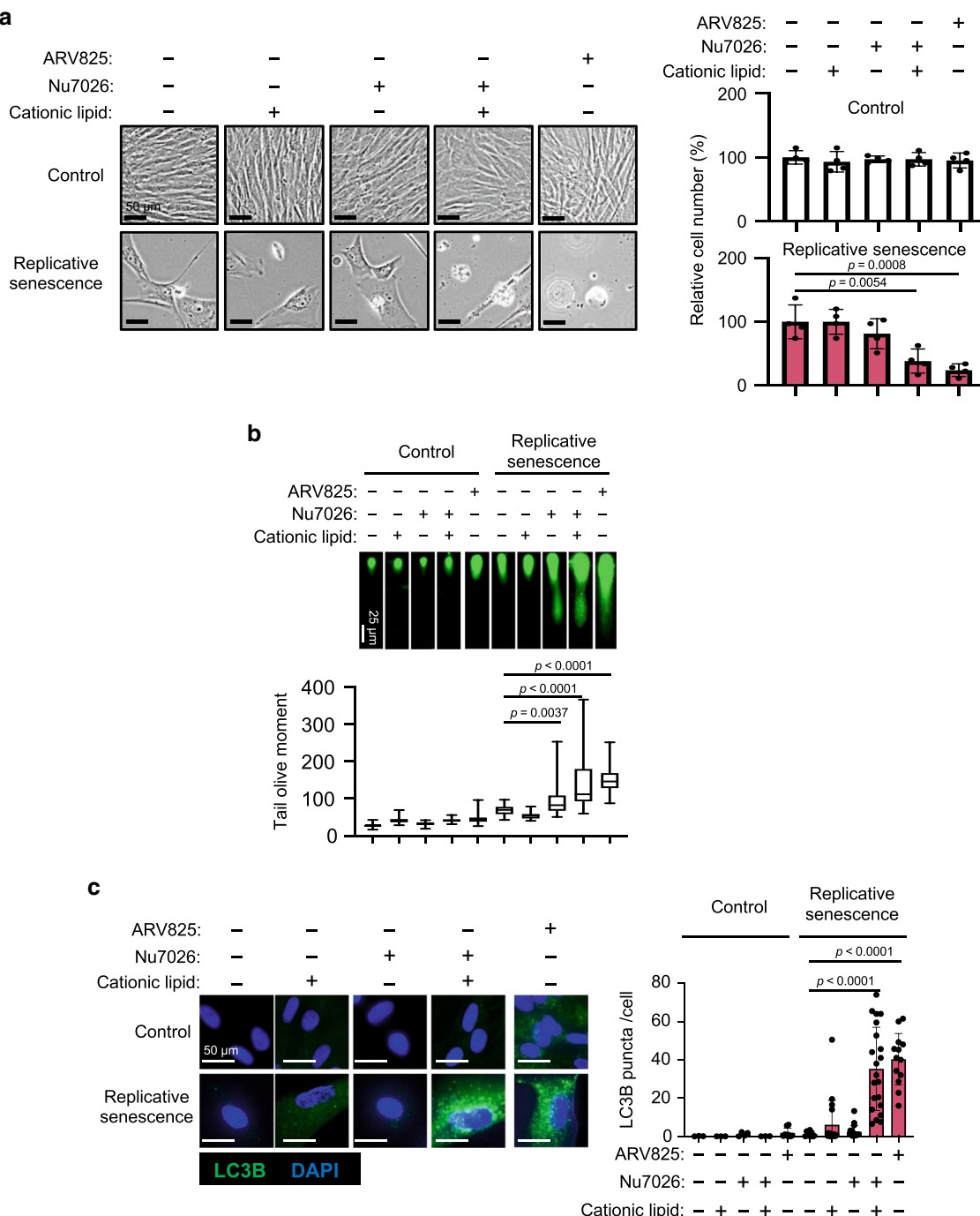

**Fig. 6 Autophagy activation is required for BETd-induced sebolysis.** Early passage (control) or late passage (replicative senescence) TIG-3 cells were treated with or without chemicals (10 nM ARV825, 20 μM Nu7026, 6 μl/ml cationic lipid) as indicated at the top of the panel, for 4 days and were then subjected to the cell proliferation analysis (**a**), neutral comet assay (**b**) or to immunofluorescence staining using the antibody against LC3B (**c**). Representative photographs of the cells in the indicated culture conditions are shown and the histogram shown at the right of the panel indicates the relative cell number (**a**). The number of LC3B puncta above threshold intensity per cells (*n* = 50) was quantified and was shown at the right of the **c**. The average tail olive moments were shown at the bottom of the **b**. Each box indicates the median and 25% and 75% quantiles, and the whiskers represent the minimum and maximum observations (**b**). Error bars indicate mean ± s.d. (*n* = 4 for **a**). The representative data from three independent experiments were shown (**a–c**). Statistical significance was determined with one-way ANOVA followed by Tukey multiple comparison test (**a**), or one-way ANOVA followed by Holm-Sidak multiple comparison test (**b–c**). *P* values < 0.05 were considered significant. Source data are provided as a Source Data file.

optimum concentration (10 nM) of ARV825 for senolysis does not provoke cell death in quiescent cells (Supplementary Fig. 3), further support the idea that BETd could be a promising senolytic drug. However, because a treatment with a high concentration (more than 50 nM) of ARV825 reduced the proliferation of

control cells (Fig. 1), it is crucial to determine the optimal concentration of ARV825 in vivo. In summary, while further studies are required to understand the precise mechanisms of BETd actions in vivo, our results have unveiled the senolytic function of a BETd, providing valuable insight into the resistance of

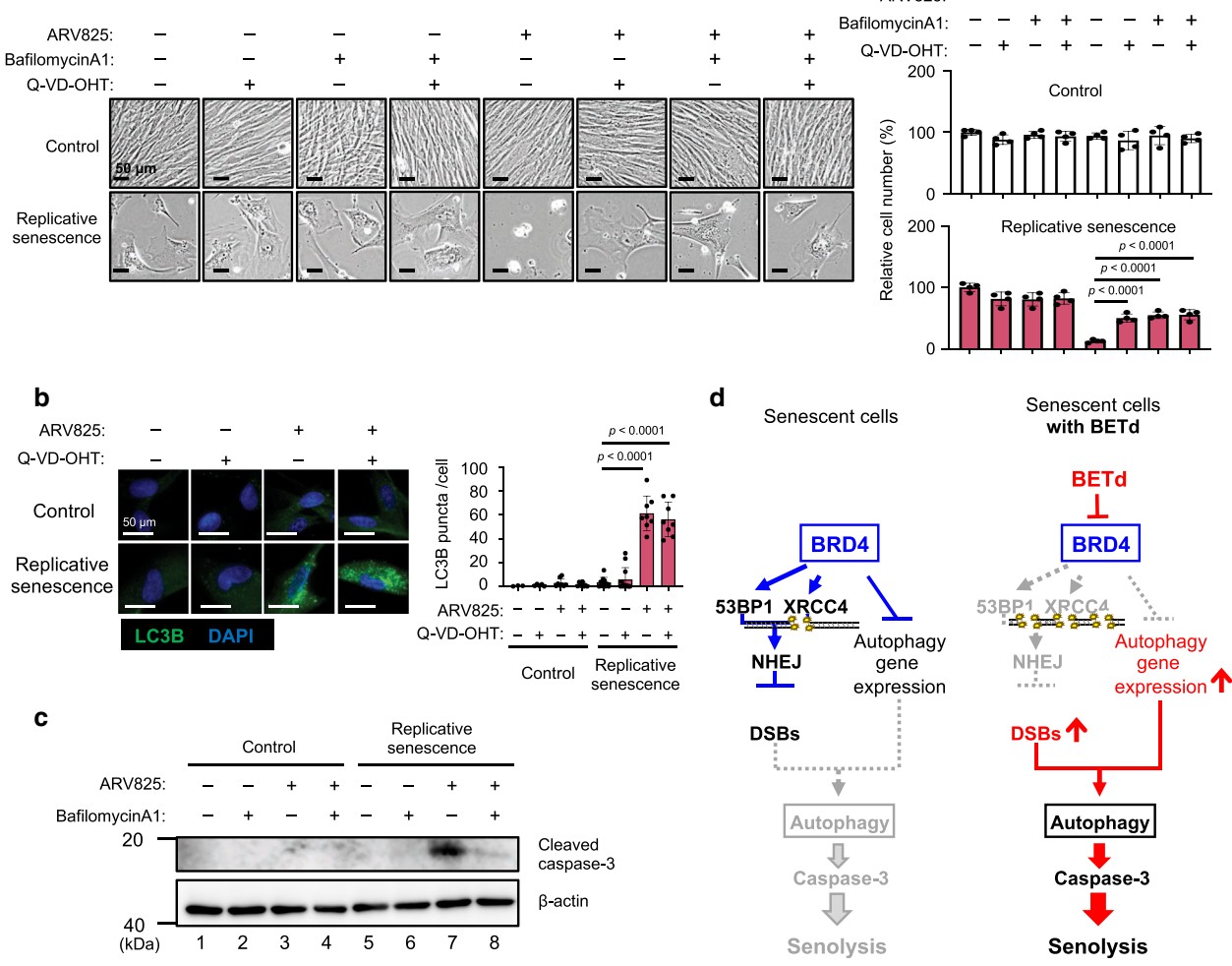

**Fig. 7 BETd provokes senolysis via NHEJ inhibition and autophagy activation. a–c** Early passage (control) or late passaged (replicative senescence) TIG3 cells were treated with or without chemicals (10 nM ARV825, 300 nM BafilomycinA1, 400 nM Q-VD-OHT), as indicated at the top of the panel, for 4 days and were then subjected to the cell proliferation analysis (**a**), immunofluorescence staining using the antibody against LC3B (**b**) or to western blotting using the antibodies shown on the right (**c**). Representative photographs of the cells in the indicated culture conditions are shown and the histogram shown at the right of the panel indicates the relative cell number (**a**). The number of LC3B puncta above threshold intensity per cells ($n = 50$) was quantified and was shown at the right of the **b**. Error bars indicate mean ± s.d ($n = 4$ for **a**) (**a**, **b**). The representative data from three independent experiments were shown (**a** to **c**). Statistical significance was determined with one-way ANOVA followed by Tukey multiple comparison test (**a**), or one-way ANOVA followed by Holm-Sidak multiple comparison test (**b**). *P* values < 0.05 were considered significant. (**d**), The model distinguishing mechanisms that may operate in the absence (left) or presence (right) of BETd in senescent cells. BRD4 protects senescent cells from autophagy-induced cell death by NHEJ activation and autophagy gene repression (left). BETd treatment, however, over-rides this steady-state and thereby causing autophagy (right). Source data are provided as a Source Data file.

senescent cells to cell death and revealing the possibilities for its control.

## Methods

**High throughput screening**. IMR90-ER:HRAS cells[44] were treated with 500 nM 4-hydroxytamoxifen to induce senescence for 7 days in T-150 flask (Corning). Then, cells were detached and seeded at 500 cells per well into the 384-well plates (Corning #3571) using Multidrop Combi (Thermo Fisher) and incubated for 24 h at 37 °C. 24 h after cell plating, compounds were added at the final 3 or 10 μM concentration and incubated for 72 h. Cell viability was monitored using CellTiter-Glo Luminescent Cell Viability assay (Promega). Luminescence signal was measured by Envision plate reader (Perkinelmer). Around 47,000 Takeda chemical library compounds were evaluated for HTS. For confirmation assay, the activity of Caspase 3 and 7 was also examined by Caspase-Glo3/7 Assay System (Promega).

**Senolytic drugs**. Senolytic drugs used are s follows: JQ1, ARV825, and ABT263 were obtained from Medchem Express (cat#: HY-13030, HY-16954, HY-10087, respectively). OTX015 and 17-DMAG were purchased from Selleck (cat#: S7360

and S1142, respectively)). Quercetin and Dasatinib were purchased from Cayman (cat#: 10005169) and LC Laboratories (cat#: D-3307), respectively.

**Cell culture**. TIG-3, hRPE, and HCT116 cells were obtained from the Japanese Cancer Research Resources Bank (JCRB), Lonza Inc., and ATCC, respectively. Mouse embryonic fibroblasts (MEFs) were established from day 13.5 mouse embryos. Mouse primary hepatic stellate cells (mHSCs) were isolated from mouse liver by the following procedure[13]. Briefly, the livers were isolated from the blood-removed 10-week-old mice. The gall bladder was removed, and then the livers were carefully excised into small pieces. The small liver pieces were incubated in PBS containing 0.05% trypsin and 0.53 mmol/L EDTA (Nacalai Tesque) for 14 minutes at 37 °C with gentle mixing at every 2 minutes. The cells were passed through 70-μm cell strainers (BD Biosciences), washed with DMEM supplemented with 10% FBS at least two times, and cultured in tissue culture dishes. After five days, α-SMA–positive cells were confirmed by cell staining and quantitative PCR analysis, and used as active hepatic stellate cells. All cells were cultured in Dulbecco's Modified Eagle's medium supplemented with 10% fetal bovine serum (FBS, Gibco, cat#: 10270). Early passage TIG-3 cells (<46 population doublings) were used as control cells, and late passage TIG-3 cells (>70 population doublings) that ceased proliferation were used as replicative senescent cells. For retroviral infection, TIG-3

cells were rendered sensitive to infection by ecotropic retroviruses[45]. Cells were infected with retroviruses encoding RasV12 (in pBabe-puro[46]), XRCC4 (in pMarX-puro[47]). After puromycin selection, pools of drug-resistant cells were analysed 10 days after infection. For chemical treatment, Nu7026 was purchased from Selleck (cat#: S2893). Bafilomycin A1 and Q-VD-OHT were purchased from Sigma (cat#: B1793 and SML0063, respectively). We have confirmed the absence of mycoplasma contamination in our tissue culture cells.

**In vitro senescence induction.** For the induction of oncogene-induced senescence, IMR-90 ER:RAS cells[44] were cultured with 4-OHT for 7days and TIG-3 cells infected with retroviruses encoding RasV12 were cultured 10 days after infection[44]. For the induction of doxorubicin-induced senescence, TIG-3, hRPE, HCT116, and MEF cells were incubated with doxorubicin (DXR, cat#: 046-21523, FUJIFILM Wako Chemicals, 250, 100, 200, and 100 ng/ml, respectively) for 10 days. mHSC cells were treated with 275 μM deoxycholic acid (DCA, cat#: 046-18811, FUJIFILM Wako Chemicals) for 10 days.

**Cell proliferation assay.** Cells were seeded in 12-well plates. The number of cells in the same spot was counted every day with All-in-One Fluorescence Microscope (BZ-710, Keyence). The relative cell number was calculated based on an adjusted cell number on day 1 set at 1.0.

**Cell counting.** Cells were stained with trypan blue after trypsin treatment. Live cells were counted using haemocytometer. The relative cell number was calculated based on the adjusted cell number in the control plate set at 1.0.

**Apoptosis assay.** Cells were stained with a fluorescein isothiocyanate-Annexin solution for 15 min at room temperature using an Apoptosis/Necrotic/Healthy Cells detection kit (Promokine). Apoptotic cells were observed and photographed with All-in-One Fluorescence Microscope (BZ-710; Keyence). The area of apoptotic positive signals in the images was calculated in an automated manner by BZ-X 700 analyzer software (Keyence).

**Plasmid.** cDNA of XRCC4 was cloned into the pMarX-puro retrovirus vector[47] using the following primers: 5′-GAGGGGATCCATGGAGAGAAAAATAAGCAG AATCCACC-3′, and 5′- TCCCGAATTCTTAAATCTCATCAAAGAGGTCTTC TGGG-3′. cDNA was sequenced on a Genetic Analyzer 3130 (Applied Biosystems) using a BigDye Terminator V3.1 Cycle Sequencing Kit (Applied Biosystems).

**RNAi.** RNAi was performed by the transfection of siRNA oligos using the Lipofectamine RNAiMAX transfection reagent (Thermo Fisher Scientific), according to the manufacturer's instructions. The sequence of siRNA oligos was as followed.

BRD2 (BRD2, Thermo Fisher Scientific, validated Silencer Select siRNA, ID: s12071). BRD3 (BRD3, Thermo Fisher Scientific, validated Silencer Select siRNA, ID: s15544[48]). BRD4_1 (BRD4, Thermo Fisher Scientific, validated Silencer Select siRNA, ID: S23901[48]). BRD4_2 (BRD4, Dharmacon, siGENOME siRNA, ID: M-004937-02-0005). BRD4_short (BRD4, Qiagen, ID: SI05044865[22]). BRD4_long (BRD4, Qiagen, ID: SI05044872[22]). XRCC4_1 (XRCC4, Sigma Aldrich, Mission siRNA, ID: SASI_Hs01_00114717). XRCC4_2 (XRCC4, Dharmacon, ON-TARGET plus siRNA, ID: L-004494-00-0005[49]). 53BP1_1 (53BP1, Sigma Aldrich, Mission siRNA, ID: SASI_Hs02_00340027). 53BP1_2 (53BP1, Dharmacon, ON-TARGET plus siRNA, ID: L-003548-00-0005[50]).

**Immunoblotting.** Proteins were extracted using with RIPA buffer with 1% Protease inhibitor cocktail (Nacalai Tesque). After determination of the protein concentration using Protein Quantification Assay (Takara Bio Inc.), all samples were denatured in Laemmli sample buffer for 5 min at 100 °C. The denatured samples were separated by SDS-polyacrylamide gel electrophoresis and transferred onto a polyvinylidene difluoride membrane (EMD Millipore). After blocking with 5% milk or 5% BSA, the membranes were incubated with the primary antibodies as follows: β-actin (1:1000, Sigma, cat#: A5316), BRD2 (1:1000, Abcam, cat#: ab139690), BRD3 (1:1000, Santa cruz, cat#: sc-81202), BRD4 (1:1000, Cell signaling, cat#: 13440), BRD4 (1:1000, abcam, cat#: 128874). Cleaved caspase3 (1:1000, Cell signaling, cat#: 9664), LaminB1 (1:1000, Abcam, cat#: ab16048), p21 (1:1000, Cell signaling, cat#: 2947 or Abcam, cat#: ab107099), XRCC4 (1:1000, Santa cruz, cat#: sc-2711087). The membranes were then incubated with the secondary antibodies (1:1000, Cell signaling) and visualized with Amersham ECL prime/select (GE Healthcare), followed by detection with chemiluminescence using LAS-3000mini imaging system (Fujifilm) and by anlaysis of data using Multi Guage V3.1 (Fujifilm). Uncropped and unprocessed scans of the blots are included in the Source Data file.

**Immunoprecipitation.** Immunoprecipitation was performed using Dynabeads Protein G (Thermo Fisher Scientific, cat#: 10004D) according to the following procedure[33]. Briefly, nuclear fractions were obtained by NE-PER nuclear extraction kit (Thermo Fisher Scientific) following the manufacturer's instructions. Nuclear samples were pre-cleared by incubation with Dynabeads protein G at 4 °C for 1 h in

IP buffer (20 mM Tris pH7.5, 150 mM NaCl, 1% Triton-X 100, and protease inhibitor (Nacalai tesque)) and then incubated with 3 μg of antibodies (53BP1, cat#: NB100-304 Novus biologicals) at 4 °C for overnight. The samples were incubated with 30 μl of Dynabeads protein G at room temperature for 2 h. Immunoprecipitated proteins were analyzed by immunoblotting as already described.

**Immunofluorescence and immunohistochemistry.** Immunofluorescence and immunohistochemistry were performed using the primary antibodies against α-SMA (1:2000, Sigma, cat#: A5228), p21 (1:50, Abcam, cat#: ab2961), 53BP1 (1:500, Santa cruz, cat#: sc-22760), IL-6 (1:400, Abcam, cat#: ab6672) and Gro-α (1:500, Abcam, cat#: ab17882) for mouse samples, and γH2AX (1:2000, Millpore, cat#: 05-321, 1:100, Abcam, cat#: ab2893), and LC3B (1:100, Cell signaling, cat#: 2775) for human samples[51]. The nuclei were counterstained with DAPI (Dojindo) or hematoxylin (Dojindo). Fluorescence and histological images were observed and photographed using All-in-One Fluorescence Microscope (BZ-710; Keyence). The area or number of positive signals in the images was calculated in an automated manner by BZ-X 700 analyzer software (Keyence).

**Quantitative real-time PCR.** Quantitative real-time PCR was performed by the following procedure[7]. Total RNA was extracted using TRIzol (Thermo Fisher Scientific) or RNeasy mini kit (Qiagen) according to the manufacture's protocol. cDNA was synthesized using a PrimeScript RT reagent kit (Takara Bio Inc.). Quantitative real-time RT-PCR was performed on a StepOnePlus PCR System (Applied Biosystems) using SYBR Premix EX Taq (Takara Bio Inc.). The mRNA expression levels of each gene were calculated relative to β-actin expression levels. The PCR primer sequences used are as follows: human *xrcc4* 5′-GCAATGGAA AAAGGGAAATATGTT-3′ (forward) and 5′-GCTGGTCCTGCTCCTGACA-3′ (reverse), human *brd2* 5′- GAGGTGTCCAATCCCAAAAAGC-3′ (forward) and 5′- ATGCGAACTGATGTTTCCACA-3′ (reverse), human *brd3* 5′- TCAAATTGA ACCTGCCGGATT-3′ (forward) and 5′- TGCATACATTCGCTTGCACTC- 3′ (reverse), human *brd4* 5′- ACCTCCAACCCTAACAAGCC -3′ (forward) and 5′- TT TCCATAGTGTCTTGAGCACC -3′ (reverse), human *malp1lc3b* 5′- AGAAG GCGCTTACAGCTCAA -3′ (forward) and 5′- AGATTGGTGTGGAGACGC TG -3′ (reverse), human *sqstm1* 5′- GCCTTGTACCCACATCTCCC -3′ (forward) and 5′- GAGAAGCCCATGGACAGCAT -3′ (reverse), human *β−actin* 5′- TGG ATCAGCAAGCAGGAGTATG -3′ (forward) and 5′- GCATTTGCGGTGGACG AT -3′ (reverse).

**RNA-sequencing analysis.** RNA isolation was performed using TRIzol. Library preparation was performed using a TruSeq stranded mRNA sample prep kit (Illumina, San Diego, CA) according to the manufacturer's instructions. Sequencing was performed on an Illumina HiSeq 3000 platform in a 75-base single-end mode. Illumina Casava ver.1.8.2 software was used for base calling. Sequenced reads were mapped to the human reference genome sequences (hg19) using TopHat ver. 2.0.13 in combination with Bowtie2 ver. 2.2.3 and SAMtools ver. 0.1.19. The number of fragments per kilobase of exon per million mapped fragments (FPKMs) was calculated using Cufflinks ver. 2.2.1.

**Neutral comet assay.** Cultured cells were harvested using trypsin/ EDTA (Nacalai tesque). In all, $1.5 \times 10^5$ cells were mixed with 1% low melting agarose gel at 1/1 ratio (v/v). The mixture was put on slide glass precoated with low melting agarose gel. After cooling down, comet slides were incubated with lysis solution (Trevigen) for 30 min at 4 °C. Neutral electrophoresis was done for 20 min at 30 volts in TBE buffer. Slides were washed with TBE buffer and 70% EtOH for 5 min. After slides were completely air dry, nuclei were stained with Midori Green Direct (Nippon Genetics Europe). Comet image was observed and photographed using All-in-One Fluorescence Microscope (BZ-710; Keyence). The olive tail moment was defined as the percentage of DNA in the tail × tail moment length and was quantified by using CometScore 2.0 software. In all experiments, the olive tail moment was calculated with more than 50 comet tails.

**Animal experiments.** C57/BL6 and Nude (nu/nu) mice were purchased from Charles River Laboratories Japan, Inc. The mice were maintained under specific pathogen-free (SPF) conditions, at 23°C ± 2 C and 55% ± 15% humidity on a 12-h light–dark cycle, and fed normal diet (ND, CE-2 from CLEA Japan Inc., composed of 12 kcal% fat, 29 kcal% protein, 59 kcal% carbohydrates) or high-fat diet (HFD, D12492 from Research Diets Inc., composed of 60 kcal% fat, 20 kcal% protein, 20 kcal% carbohydrates) ad libitum. All mouse experiments were approved by the Animal Research Committee of Research Institute for Microbial Diseases, Osaka University.

**Chemically induced hepatocarcinoma.** Chemically induced hepatocacrinoma was performed by the following procedure[13]. Briefly, 50 μl of a solution 0.5% (w/v) DMBA (7,12-dimethylbenz [a] anthracene, Sigma) in acetone was applied to mouse dorsal surface on postnatal day 4–5. From the age of 4 weeks old, pups were weaned and continuously fed HFD or ND. At the age of 16weeks, the intraperitoneal administration with ARV825 (5 mg/kg weight) or vehicle was started using mice with over 45 g of body weight. ARV825 or vehicle was administrated daily for

5 consecutive days over 2 weeks followed with a one-week interval. This protocol was continued until euthanized. Tumor size was classified as followed with 1–3 mm, 3–6 mm, >6 mm by measuring the visible tumors and tumor number was counted.

**Xenograft**. $1\times10^6$ HCT116 cells were mixed with Matrigel (BD Biosciences) at 1/1 ratio (v/v) and subcutaneously injected into Nude mice (4–5 weeks old). After 1week, the mice were treated with DXR at 1 mg/kg weight every day for 1 week. 1 week after DXR treatment, ARV825 (5 mg/kg weight) or vehicle was administrated intraperitoneally 10 times in 2 weeks. Tumor size was measured with Vernier caliper. Tumor volume (mm³) was calculated as (length × width × width)/2[52].

**Statistics and reproducibility**. Statistical significance was determined with two-tailed unpaired Student's *t*-test (Figs. 2b and 5b, Supplementary Figs. 4a, 6, 8a, 9a, 11), one-way analysis of variance followed by Holm-Sidak multiple comparison test (Figs. 2c, 4b, 4c, 5e, 5f, 6b, 6c, 7b, Supplementary Fig. 3b, 4c, 7a, 7b, 8d, 8f, 9e, 9f, 10b, 10d, 12, 13b), one-way analysis of variance followed by Tukey multiple comparison test (Figs. 3e, 5d, 6a, 7a, Supplementary Figs. 5c, 8c, 9c, 10f, 13a), or Kruskal–Wallis followed by Dunn's multiple comparison test (Fig. 3d). *P* values < 0.05 were considered significant. All experiments except Fig. 1a was repeated at least twice independently with similar results.

**Reporting summary**. Further information on research design is available in the Nature Research Reporting Summary linked to this article.

## Data availability

RNA sequencing data of Fig. 4a has been deposited in the Gene Expression Omnibus under accession number GSE140961. The source data underlying Figs. 1b–e, 2b, c, 3a, b, d, e, 4b, c, 5a–f, 6a–c, 7a–c and Supplementary Figs. 1, 2, 3b, 4, 5, 6, 7, 8, 9, 10, 11, 12, and 13 are provided as a Source Data file. Uncropped and unprocessed scans of the blots are included in the Source Data file. The remaining data are available in the article, Supplementary Information files or available from the authors upon reasonable request.

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

# ARTICLE

51. Ohtani, N. et al. Visualizing the dynamics of p21$^{Waf1/Cip1}$ cyclin-dependent kinase inhibitor expression in living animals. *Proc. Natl Acad. Sci. USA* **104**, 15034–15039 (2007).
52. Jensen, M. M., Jorgensen, J. T., Binderup, T. & Kjaer, A. Tumor volume in subcutaneous mouse xenografts measured by microCT is more accurate and reproducible than determined by 18F-FDG-microPET or external caliper. *BMC Med. Imaging* **8**, 16 (2008).

## Acknowledgements

We thank Drs. T. Henta, N. Tarui, and S. Sato (Takeda Pharmaceutical Company Ltd.) for RINGO-T project and D. Okuzaki (RIMD, Osaka University, Japan) for performing RNA-seq analysis. We also thank Ms. T. Kondo and M. Suzuki (RIMD, Osaka University, Japan) for technical support throughout this study. We are grateful to members of Hara's laboratory for helpful discussion during the preparation of this manuscript. This work was supported in part by grants from Japan Agency of Medical Research and Development (AMED) under grant number 19gm5010001h0003 and 19cm0106401h0004 and from the Japan Society for the Promotion of Science (JSPS) under grant number 18K15065. M. Wakita was supported by the Taniguchi Memorial Fellowship Program.

## Author contributions

E.H. and M.W. designed the experiments. M.W. performed most of the experiments. A.T., O.S., Y.I, and H.I. performed high-throughput screening. T.M.L. performed immunostaining experiments. M.N. provided senescent cells. M.W., T.M., S.K., N.O., T.Y., and E.H. analysed the data. E.H. wrote the paper and oversaw the projects. All the authors discussed the results and commented on the paper.

## Competing interests

The authors declare no competing interests.
