## [Peer Review File · Nature Communications]

Reviewers' comments:

Reviewer #1 (Remarks to the Author): Expert in senescence

In this manuscript entitled "BET inhibitor provokes senolysis by exploiting cellular vulnerability of senescent cells" Wakita et al., by performing an unbiased high-throughput screening of chemical compound libraries and bio-functional assays, have identified bromodomain and extra-terminal domain (BET) family protein inhibitors (BETi) as a novel class of senotherapy drugs. The authors show that the blockade of BRD4 by chemical inhibition and RNA interference promotes apoptosis in senescent cell. Mechanistically Wakita and colleagues provide evidence that this process involves the integration of two processes: the exacerbation of DNA double-strand breaks by blocking non-homologous end joining repair and the activation of the autophagy machinery. They further validate their results by using (i) a mouse model of obesity-associated liver cancer development, where the use of BETi treatment reduces the accumulation of senescent hepatic stellate cells and results in reduced tumour burden, and (ii) a xenograft model of chemotherapy-induced senescent, providing evidence of the synergistic effects of senogenic drugs in combination with senotherapies.

Overall, the experiments are well designed and conducted, and the results support the conclusions. Also, the manuscript is interesting, timely and well written. However, I have some questions that should be addressed before it can be recommended for publication:

1. The data of the Extended Figure 3 suggest that ARV825 treatment does not induce apoptosis in quiescent TIG-3 cells, whereas ABT-263 appears to do it (as per the Annexin V levels). However, the Fig. 1b and the Extended Data Fig. 1 show that ABT-263 is the only senolytic not affecting the relative control cell numbers, even at the highest concentration, whereas ARV825 significantly reduces control cell numbers at 50 nM and 1 μ M. This indicates a more specific senolytic activity for ABT-263. How can the authors explain these results with the same cellular type? I think conclusions should be taken with care.

2. In the experiment of doxorubicin-induced senescence by using mice bearing subcutaneous tumours with HCT116 cells I wonder whether, in addition to the reduction of p21 and 53BP1 cells (Fig. 2h), the use of ARV825 results in increased levels of apoptosis. The histology data would be more solid with data on CC3 or TUNEL, and Ki67.

3. I am not convinced of the idea that ARV825 partially provokes senolysis by inhibiting the *xrcc4* gene expression and would favour to include these data in the Supplementary Material. The fact that the overexpression of XRCC4 failed to rescue the ARV825-induced DSBs and senolysis suggests a bystander effect rather than causal role. Figures 2h and 3b show that ARV825 treatment increase γ H2AX and reduce the 53BP1 levels/foci in chemotherapy-induced senescence and replicative senescence, indicating that the damage is not efficiently repaired and resulting in reduced cell numbers (as in Fig. 3g). However, in the Extended Data Fig. 7 the siRNA-mediated downregulation of XRCC4 increases γ H2AX foci but also 53BP1 in senescent cells, indicating active NHEJ repair. Can the authors comment on this?

4. Irrespectively of XRCC4, the proposed role of DSBs and autophagy in BETi-induced senolysis seems more solid to me and well supported by the in vitro approaches. Despite these exciting data the absence of some in vivo support in at least one of the experimental models tempers my enthusiasm. Following my comments in point 2, I find crucial to provide in vivo evidence that ARV825 treatment results in increased damage (γ H2AX levels) and autophagy (LC3 and/or p62 levels). In addition, the combined use of the autophagy inhibitors chloroquine or leupeptin in combination with ARV825 in mice could provide conclusive assessment of the autophagy-induced apoptosis claimed by the authors. Alternatively, and accordingly with the proposed model, I wonder whether starvation (48h), a well-known method to widely induce autophagy in mice, in combination with ARV825, increases apoptosis in the tumours.

Reviewer #2 (Remarks to the Author): Expert in mouse models of liver cancer

"BET inhibitor provokes senolysis by exploiting cellular vulnerability of senescent cells", by Wakita et al.

The authors present their results based on high-throughput screening to discover senolytic compounds, and show data on BET inhibitor (BETi) as a promising new drug candidate for senolysis. They have demonstrated the effectiveness of the BETi in inducing death of senescent cells both in culture (senescence induced by various means), as well as in two in vivo models, demonstrating the effects of senolysis resulting in reduced tumor growth. They then go on to uncover the mechanistic basis of BETi's senolytic action, and in a series of experiments, have demonstrated convincingly and elegantly that the interplay of DNA double strand breaks (DSB) and autophagy are important mediators of the BETi senolytic effect. The step-by-step detailed approach taken to uncover the proof the role of DSB and autophagy is commendable.

The experiments are done very well and well controlled, and the results clearly demonstrate the effects of BETi and its mode of action.

Specific points:

1. The results of the HTS should be shown (as a table). While BETi was chosen for further study, it is unclear if it is the top ranking hit, or if other compounds from various classes have also been identified. Moreover, whether currently available senolytics or their similars were also identified from the screen needs to be clarified.

2. The authors have narrowed down the effects of BETi to BRD4, which when silenced, phenocopies the BETi effect. BRD4 comes in 4 isoforms, which have been shown to have similar and opposing effects, at least with respect to tumorigenesis. Hence, it is important to decipher which of the 4 BRD4 isoforms would be involved in the senolytic effect, by specifically knocking down each of them.

Overall, this is a very interesting manuscript detailing a new role for BETi in senolysis. While not translatable to the clinical setting yet, it provides new information which could be used to design and develop better classes of senolytics.

Reviewer #3 (Remarks to the Author): Expert in BET inhibitors

Masahiro W et al demonstrated that BET inhibitors (BETi) could be used as therapeutic drugs to eliminate senescent cells by inducing autophagy-associated apoptosis (senolysis). The authors started with treating Ras-induced senescent cells using an unbiased high-throughput screening of chemical compound libraries including 47,000 small molecules and demonstrated that four BETi significantly induced apoptosis. Furthermore, they used another newly-developed BETi named ARV-825 to investigate the underlying mechanisms and found that ARV-825 inhibited NHEJ repair by downregulating XRCC4 gene expression. The undermined DNA repair accumulated DSBs, and subsequently provoked autophagy gene expression and apoptosis.

Overall, this study applied a state-of-the-art method to identify BETi as effective drugs to induce senolysis in senescent cells, which is solid. However, the description of the results and experimental designs might have some space to improve. Also, a thorough discussion about the contribution of this study in the field and the emphasis of the novelty might be beneficial for the whole story.

Major concerns:

1. A study has previously demonstrated that ARV-825 can effectively suppress c-MYC levels,

resulting in apoptosis (PMID: 26051217). Although in this present study, the authors have demonstrated that this drug cannot affect c-MYC expression in senescent cells by RNA-Seq analysis, the further experiments such as RT-qPCR and western blot analysis in different cell lines might be useful to confirm that c-MYC is not altered in the systems the authors analyzed.

2. While the authors have mentioned that 15 small molecules induced cell death in Ras-induced senescent cells, except four BET inhibitors, the authors did not list or describe the other molecules.

3. Fig. 2b, the statistical analysis is missing between two groups.

4. Fig. 2c about α SMA and Gro α staining is not well-described in the manuscript. Seemingly, this panel did not answer any questions. A brief introduction might be needed to talk about α SMA and Gro α .

5. Fig. 2g, a tumor graph showing tumors from different treatment groups is missing.

6. Fig. 2h and 3b, what is the rationale to detect 53BP1 here? In the manuscript, the authors officially introduced 53BP1 from Fig.3f, but in these two figures, they have shown some results without describing them.

7. A previous study has revealed that BETi can block IR-induced recruitment of XRCC4 to the chromatin (PMID29346775). How to account for the possibility that BETi also can physically block the access of XRCC4 to NHEJ repair instead of downregulating its expression?

8. Although the authors have mentioned that "NHEJ, but not homologous recombination (HR), is the major DNA repair mechanism for DSBs in non-dividing cells", it would be interesting to experimentally detect whether ARV-825 affects HR.

9. For all the figures showing senescent cells, it would be good to show the β -gal staining as well.

10. For siRNA knockdown, the efficiency of siRNA knockdown should be confirmed at the protein level by western blot analysis.

11. Based on Fig. 3b, 3h, 53BP1 protein level is enhanced in senescent cells. A discussion about this phenomenon should be included in the manuscript.

Minor issues

1. Please check the spellings and grammar in the manuscript.
2. The values of scale bars are missing in some figures.
3. Please check the labelling alignment.
4. The right panel in Fig. 4a requires group names.
5. In the figure legends, please describe the statistical analysis.

Point-by-point responses to the reviewers' comments

We sincerely thank both the reviewers and the editors for the constructive and thoughtful reviews provided for our manuscript. We are grateful for their shared appreciation of our manuscript as “*Overall, the experiments are well designed and conducted, and the results support the conclusions. Also, the manuscript is interesting, timely and well written. (Reviewer 1)*”, “*Overall, this is a very interesting manuscript detailing a new role for BETi in senolysis. While not translatable to the c clinical setting yet, it provides new information which could be used to design and develop better classes of senolytics.(Reviewer 2)*”, and “*Overall, this study applied a state-of-the-art method to identify BETi as effective drugs to induce senolysis in senescent cells, which is solid. (Reviewer 3)*”. All of the comments raised are quite helpful to improve our manuscript. We believe that by addressing the reviewers' concerns we have produced a more solid and interesting manuscript. Our point-by-point responses to the reviewers' comments are detailed below, with the original comments bolded. Furthermore, in order to describe the outline of our paper more concretely, we have changed the title of our paper as follows: “A selective BET family protein degrader provokes senolysis by targeting NHEJ and autophagy in senescent cells.”

Reviewer #1 (Remarks to the Author): Expert in senescence

In this manuscript entitled "BET inhibitor provokes senolysis by exploiting cellular vulnerability of senescent cells" Wakita et al., by performing an unbiased high-throughput screening of chemical compound libraries and bio-functional assays, have identified bromodomain and extra-terminal domain (BET) family protein inhibitors (BETi) as a novel class of senotherapy drugs. The authors show that the blockade of BRD4 by chemical inhibition and RNA interference promotes apoptosis in senescent cell. Mechanistically Wakita and colleagues provide evidence that this process involves the integration of two processes: the exacerbation of DNA double-strand breaks by blocking non-homologous end joining repair and the activation of the autophagy machinery. They further validate their results by using (i) a mouse model of obesity-associated liver cancer development, where the use of BETi treatment reduces the accumulation of senescent hepatic stellate cells and results in reduced tumour burden, and (ii) a xenograft model of chemotherapy-induced senescent, providing evidence of the synergistic effects of senogenic drugs in combination with senotherapies.

Overall, the experiments are well designed and conducted, and the results support the conclusions. Also, the manuscript is interesting, timely and well written. However, I have some questions that should be addressed before it can be recommended for publication:

1. The data of the Extended Figure 3 suggest that ARV825 treatment does not induce apoptosis in quiescent TIG-3 cells, whereas ABT-263 appears to do it (as per the Annexin V levels). However, the Fig. 1b and the Extended Data Fig. 1 show that ABT-263 is the only senolytic not affecting the relative control cell numbers, even at the highest concentration, whereas ARV825 significantly reduces control cell numbers at 50 nM and 1 uM. This indicates a more specific senolytic activity for ABT-263. How can the authors explain these results with the same cellular type? I think conclusions should be taken with care.

Response:

We thank the reviewer for pointing this out. As the reviewer mentioned, a treatment with a high concentration (more than 50 nM) of ARV825 reduced the proliferation of control cells. However, this was not the case when the control cells were treated with the optimum concentration (5~25 nM) of ARV825 for senolysis (Fig. 1). Moreover, unlike previously reported senolytic drugs, ARV825 exhibits robust senolytic activity even at nanomolar concentrations (5~10 nM). These results, together with the observation that the optimum concentration (10 nM) of ARV825 for senolysis did not provoke cell death in quiescent cells (Supplementary Fig. 3), further support the idea that a BET degrader such as ARV825 could be a promising senolytic drug, depending on the concentration. However, this dependence on the concentration of the BET degrader is quite important. Thus, in line with the reviewer's comment, we have carefully discussed this point in the revised text on page 14, lines 10 to 12.

It is worthwhile to note that ABT263 targets both Bcl-2 and Bcl-xL, and Bcl2/Bcl-xl reportedly play crucial roles also in quiescent cells (Janumyan et al., JBC, 283, 34108-34120, 2008). Therefore, it is not surprising that ABT263 treatment affects not only senescent cells but also quiescent cells.

2. In the experiment of doxorubicin-induced senescence by using mice bearing subcutaneous tumours with HCT116 cells I wonder whether, in addition to the reduction of p21 and 53BP1 cells (Fig. 2h), the use of ARV825 results in increased levels of apoptosis. The histology data would be more solid with data on CC3 or TUNEL, and Ki67.

Response:

We thank the reviewer for this helpful comment. In line with the reviewer's comment,

we have examined the levels of Cleaved caspase 3 (an apoptosis marker) and found that the levels of Cleaved caspase 3 staining were substantially increased by co-treatment with DXR and ARV825 in xenografted tumour tissues. These results are now included in the new Fig. 2h and described in the revised text on page 11, lines 15 to 18.

3. I am not convinced of the idea that ARV825 partially provokes senolysis by inhibiting the xrcc4 gene expression and would favour to include these data in the Supplementary Material. The fact that the overexpression of XRCC4 failed to rescue the ARV825-induced DSBs and senolysis suggests a bystander effect rather than causal role. Figures 2h and 3b show that ARV825 treatment increase gH2AX and reduce the 53BP1 levels/foci in chemotherapy-induced senescence and replicative senescence, indicating that the damage is not efficiently repaired and resulting in reduced cell numbers (as in Fig. 3g). However, in the Extended Data Fig. 7 the siRNA-mediated downregulation of XRCC4 increases gH2AX foci but also 53BP1 in senescent cells, indicating active NHEJ repair. Can the authors comment on this?

Response:

As the reviewer pointed out, overexpression of XRCC4 failed to rescue the ARV825-induced DSBs and senolysis in senescent cells. However, please note that ARV825 not only reduces XRCC4 expression but also inhibits the interaction between BDR4 and 53BP1 (an essential step for 53BP1 foci formation) (Fig. 3b and d). Moreover, although XRCC4 reportedly activates DNA ligase IV (Polo & Jackson, Genes Dev. 55, 97-110, 2014), there are no reports showing that XRCC4 promotes the 53BP1 foci formation. Indeed, we observed that the overexpression of XRCC4 failed to rescue 53BP1 foci formation in senescent cells treated with ARV825 (new Supplementary Fig. 10e and f), indicating that XRCC4 alone does not have any impact on the recruitment of 53BP1 to the DNA break region. Therefore, it is quite reasonable that the overexpression of XRCC4 failed to rescue ARV825-induced DSBs and senolysis in senescent cells (new Supplementary Fig. 10). On the other hand, importantly, the siRNA-mediated downregulation of XRCC4 provoked the exacerbation of DSBs (a sign of NHEJ inhibition) in senescent cells (Supplementary Fig. 8d). Taken together, it is most likely that although the overexpression of XRCC4 alone is not sufficient to rescue ARV825-induced NHEJ inhibition in senescent cells, the expression of XRCC4 is essential for activating the NHEJ repair machinery.

Along similar lines, because XRCC4 does not have any impact on 53BP1 foci formation (new Supplementary Fig. 10e and f), it is not surprising that the down-regulation of XRCC4 inhibits the NHEJ repair without blocking the 53BP1 foci formation in senescent cells (new Supplementary Fig. 8d to f). We have discussed these points in the revised text on page 9, line 16 to page 10, line 10. Moreover, in accordance with the reviewer's suggestion, we have moved the XRCC4 knock-down data to the new Supplementary Figure 8.

4. Irrespectively of XRCC4, the proposed role of DSBs and autophagy in BETi-induced senolysis seems more solid to me and well supported by the *in vitro* approaches. Despite these exciting data the absence of some *in vivo* support in at least one of the experimental models tempers my enthusiasm. Following my comments in point 2, I find crucial to provide *in vivo* evidence that ARV825 treatment results in increased damage (γH2AX levels) and autophagy (LC3 and/or p62 levels). In addition, the combined use of the autophagy inhibitors chloroquine or leupeptin in combination with ARV825 in mice could provide conclusive assessment of the autophagy-induced apoptosis claimed by the authors. Alternatively, and accordingly with the proposed model, I wonder whether starvation (48h), a well-known method to widely induce autophagy in mice, in combination with ARV825, increases apoptosis in the tumours.

Response:

We thank the reviewer for the insightful comments. Accordingly, we performed the suggested experiments using antibodies against γH2AX (DNA damage marker) or LC3B (autophagy marker), and found that the levels of γH2AX staining and LC3B staining were substantially increased in tumour tissues co-treated with DXR and ARV825. These results are now included in the new Fig. 2h and described in the revised text on page 11, lines 15 to 18.

In accordance with the reviewer's suggestion, we have also examined the effects of the combined use of the autophagy inhibitor chloroquine with ARV825 *in vivo* (HCT116 xenograft mice). As the reviewer predicted, the combined use of chloroquine with ARV825 significantly attenuated the tumour suppressive effect of ARV825 in HCT116 xenograft mice (new Supplementary Figure 13b). These results, together with the observation that the combined use of chloroquine with ARV825 attenuates the ARV825-induced senolysis in cultured cells (new Supplementary Figure 13a), strongly suggest that ARV825 provokes senolysis at least partly through activating the autophagic pathway both *in vitro* and *in vivo*. These results are described in the revised text on page 11, lines 12 to 14 and lines 18 to 20.

Reviewer #2 (Remarks to the Author): Expert in mouse models of liver cancer

“BET inhibitor provokes senolysis by exploiting cellular vulnerability of senescent cells”, by Wakita et al.

The authors present their results based on high-throughput screening to discover senolytic compounds, and show data on BET inhibitor (BETi) as a promising new drug candidate for senolysis. They have demonstrated the effectiveness of the BETi in inducing death of senescent cells both in culture (senescence induced by various means), as well as in two in vivo models, demonstrating the effects of senolysis resulting in reduced tumor growth. They then go on to uncover the mechanistic basis of BETi’s senolytic action, and in a series of experiments, have demonstrated convincingly and elegantly that the interplay of DNA double strand breaks (DSB) and autophagy are important mediators of the BETi senolytic effect. The step-by-step detailed approach taken to uncover the proof the role of DSB and autophagy is commendable. The experiments are done very well and well controlled, and the results clearly demonstrate the effects of BETi and its mode of action.

Specific points:

1. The results of the HTS should be shown (as a table). While BETi was chosen for further study, it is unclear if it is the top ranking hit, or if other compounds from various classes have also been identified. Moreover, whether currently available senolytics or their similars were also identified from the screen needs to be clarified.

Response:

We deeply apologize for not mentioning other hit compounds. Unexpectedly, we were unable to find any previously reported senolytic compounds in our HTS. However, 4 out of 15 top-ranking hits were BETi. Therefore, we focused on BETi in this study. We have described these points in the revised text on page 5, lines 12 to 16.

2. The authors have narrowed down the effects of BETi to BRD4, which when silenced, phenocopies the BETi effect. BRD4 comes in 4 isoforms, which have been

shown to have similar and opposing effects, at least with respect to tumorigenesis. Hence, it is important to decipher which of the 4 BRD4 isoforms would be involved in the senolytic effect, by specifically knocking down each of them.

Response-2:

We thank the reviewer for suggesting this important experiment. As the reviewer mentioned, there are 4 isoforms of human BRD4 (see the figure below). However, since the BRD4-NUT fusion protein is expressed only in NUT midline carcinoma cells, due to translocation of the *BRD4* gene to the locus encoding NUT, three BRD4 isoforms are potentially expressed in senescent cells.

As shown in the western blot in the new Supplementary Fig. 5a, senescent TIG-3 cells express the BRD4-A (long) and BRD4-C (short) isoforms, but not the BRD4-B isoform. Since the ARV825 treatment and siRNA oligos used in Extended Data Figures 4 and 8 (→new Supplementary Figs. 4 and 9) reduced both the long and short isoforms of BRD4 (see new Supplementary Fig. 5a), we examined the effects of the isoform-specific knock-down of BRD4 using the siRNA oligos described in Sakamaki *et al.* (Mol. Cell, 2017). The knock-down of the long isoform, but not the short isoform, of BRD4 caused senolysis, indicating that the long isoform plays a more important role in protecting senescent cells from senolysis (new Supplementary Figure 5b and c). These results are included in the new Supplementary Figure 5 and described in the text on page 8, lines 5 to 12.

(Adapted from Sakamaki *et al.*, (2017) Mol. Cell 66: 517-532.)

Overall, this is a very interesting manuscript detailing a new role for BETi in senolysis. While not translatable to the clinical setting yet, it provides new information which could be used to design and develop better classes of senolytics.

Response:

We thank the reviewer for such a positive comment.

Reviewer #3 (Remarks to the Author): Expert in BET inhibitors

Masahiro W et al demonstrated that BET inhibitors (BETi) could be used as therapeutic drugs to eliminate senescent cells by inducing autophagy-associated apoptosis (senolysis). The authors started with treating Ras-induced senescent cells using an unbiased high-throughput screening of chemical compound libraries including 47,000 small molecules and demonstrated that four BETi significantly induced apoptosis. Furthermore, they used another newly-developed BETi named ARV-825 to investigate the underlying mechanisms and found that ARV-825 inhibited NHEJ repair by downregulating XRCC4 gene expression. The undermined DNA repair accumulated DSBs, and subsequently provoked autophagy gene expression and apoptosis.

Overall, this study applied a state-of-the-art method to identify BETi as effective drugs to induce senolysis in senescent cells, which is solid. However, the description of the results and experimental designs might have some space to improve. Also, a thorough discussion about the contribution of this study in the field and the emphasis of the novelty might be beneficial for the whole story.

Major concerns:

1. A study has previously demonstrated that ARV-825 can effectively suppress c-MYC levels, resulting in apoptosis (PMID: 26051217). Although in this present study, the authors have demonstrated that this drug cannot affect c-MYC expression in senescent cells by RNA-Seq analysis, the further experiments such as RT-qPCR and western blot analysis in different cell lines might be useful to confirm that c-MYC is not altered in the systems the authors analyzed.

Response:

We thank the reviewer for pointing this out. As shown in the Figure below, ARV825 can suppress the levels of c-myc expression in proliferating cancer cells. Thus, we are in agreement with the previous finding that ARV825 inhibits c-myc expression in proliferating cancer cells. However, it is well known that the expression of many oncogenes including c-myc is silenced by multiple mechanisms in senescent cells (Chan & Narita, Genes Dev. 2019; Guney & Sedivy, Cell Cycle 2006). Thus, it is not

surprising that the expression of *c-myc* is no longer repressed by ARV825 treatment in senescent cells.

Proliferating HCT116 cells or TIG3 cells treated with 250 ng/ml doxorubicin (DXR) for 10 days were treated with (+) or without (-) 10 nM ARV825 for 2 days. These cells were then subjected to RT-qPCR analysis to measure the levels of *c-myc* gene expression. Error bars indicate \pm s.d. Statistical significance was determined with a one-way ANOVA followed by Dunnett's multiple comparison. *P* values <0.05 were considered significant (**<0.01).

2. While the authors have mentioned that 15 small molecules induced cell death in Ras-induced senescent cells, except four BET inhibitors, the authors did not list or describe the other molecules.

Response:

We deeply apologize for not mentioning other hit compounds. Unexpectedly, we were unable to find previously reported senolytic compounds in our HTS. However, 4 out of 15 top-ranking hits were BETi. Therefore, we focused on BETi in this study. We have described these points in the revised text on page 5, lines 12 to 16.

3. Fig. 2b, the statistical analysis is missing between two groups.

Response:

We apologize for this omission. In accordance with the reviewer's suggestion, we have performed statistical analyses in the new Fig. 2b.

4. Fig. 2c about α SMA and Gro α staining is not well-described in the manuscript.

Seemingly, this panel did not answer any questions. A brief introduction might be needed to talk about α SMA and Gro α .

Response:

We thank the reviewer for pointing this out. We agree with the reviewer on this point and have adjusted the text accordingly (page 7, lines 3 and 4).

5. Fig. 2g, a tumor graph showing tumors from different treatment groups is missing.

Response:

We have presented the tumour graph in Fig. 2g.

6. Fig. 2h and 3b, what is the rationale to detect 53BP1 here? In the manuscript, the authors officially introduced 53BP1 from Fig.3f, but in these two figures, they have shown some results without describing them.

Response:

We deeply apologize for our poor description. In accordance with the reviewer's suggestion, we have mentioned that the formation of 53BP1 foci is a marker of the DNA damage response associated with cellular senescence (see page 6, lines 13 to 16).

7. A previous study has revealed that BETi can block IR-induced recruitment of XRCC4 to the chromatin (PMID29346775). How to account for the possibility that BETi also can physically block the access of XRCC4 to NHEJ repair instead of downregulating its expression?

Response:

Because the levels of XRCC4 expression are reduced by ARV825 treatment, we have attempted to address this question by using senescent cells overexpressing XRCC4 described in Supplementary Fig. 10a. As shown in the Figure below, ARV825 treatment did not affect the recruitment of XRCC4 to the chromatin, precluding the

possibility that ARV825 can also physically block the access of XRCC4 to NHEJ repair, instead of downregulating its expression.

TIG3 cells were rendered senescent by a treatment with 250 ng/ml doxorubicin (DXR) for 10 days. These cells were then treated with or without 10 nM ARV825 for 2 days. Chromatin fractions were prepared as described previously (PMID:29346775) and were subjected to western blotting analyses using the antibodies shown on the right (lanes 1 and 2).

8. Although the authors have mentioned that “NHEJ, but not homologous recombination (HR), is the major DNA repair mechanism for DSBs in non-dividing cells”, it would be interesting to experimentally detect whether ARV-825 affects HR.

Response:

We thank the reviewer for the suggestion. However, as reported previously (Mao *et al.*, PNAS 11800-11805, 2012), the HR activity was hardly detectable in senescent HDFs (see the Figure below). Thus, it is unlikely that ARV825 mainly targets HR in senescent cells.

Homologous recombination efficiency was measured using a qPCR-based HR assay kit, according to the manufacturer's instructions (Norgen Biotek Corp cat#: 35600). Briefly, early-passage (control proliferating) and late-passage (replicative senescent) TIG-3 cells were co-transfected with the dl-1 and dl-2 plasmids. Twenty-four hrs later, cellular DNA was extracted from these cells and HR efficiency was measured by qPCR. Error bars indicate the \pm s.d. Statistical significance was determined with a Student's t-test. *P* values <0.05 were considered significant (**P <0.001).

9. For all the figures showing senescent cells, it would be good to show the β -gal staining as well.

Response:

We thank the reviewer for this suggestion. However, importantly, our group and others have reported that the SA- β -gal activity is also induced in quiescent cells and several cancer cell lines (Imai *et al.*, Cell Rep. 2014; Cristofalo, Exp. Gerontol. 2004; Dimri *et al.*, PNAS 1995). Moreover, the SA- β -gal activity is known to be dispensable for the implementation of cellular senescence (Lee *et al.*, Aging Cell, 5, 187-195, 2006). These results strongly suggest that although SA- β -gal is regarded as the gold standard for identifying senescent cells, this clearly needs to be interpreted with caution. Therefore, we are reluctant to use the SA- β -gal activity as a senescence marker. Instead, we have used multiple markers known to play key roles in the establishment of the senescent state, such as DNA damage foci, increased levels of p21 expression, and the reduction of Lamin B1 expression, throughout this study. We have discussed this point in the revised text on page 6, lines 10 to 16.

10. For siRNA knockdown, the efficiency of siRNA knockdown should be confirmed at the protein level by western blot analysis.

Response:

We are grateful for the reviewer for this comment. We have now included western blotting data in the new Figure 3f and new Supplementary Figures 4b, 8b, and 9b.

11. Based on Fig. 3b, 3h, 53BP1 protein level is enhanced in senescent cells. A discussion about this phenomenon should be included in the manuscript.

Response:

We thank the reviewer for this comment. However, it has already been reported that the formation of 53BP1 foci is increased in response to DNA damage (Imai *et al.*, Cell Rep. 2014). Moreover, there are reports showing that the 53BP1 protein level is increased in senescent cells (Benkafadar *et al.* Mol. Neurobiol. 2019). Therefore, we do not think it is worthwhile to discuss this point in this paper.

Minor issues

1. Please check the spellings and grammar in the manuscript.

Response:

The manuscript is now proofed by a native English speaker to correct spellings and grammatical errors as suggested.

2. The values of scale bars are missing in some figures.

Response:

We thank the reviewer for pointing this out and have adjusted the Figures accordingly.

3. Please check the labelling alignment.

Response:

We thank the reviewer for pointing this out. We have adjusted the Figures accordingly.

4. The right panel in Fig. 4a requires group names.

Response:

We apologize for this omission, and have adjusted Fig. 4a accordingly.

5. In the figure legends, please describe the statistical analysis.

Response:

In line with the reviewer's suggestion, we have adjusted the Figure legends accordingly.

REVIEWERS' COMMENTS:

Reviewer #1 (Remarks to the Author):

I am pleased with the authors comments and additional experiments as they have addressed the majority of my original concerns.

I would only consider to add Supplementary Fig. 13 to the main text (in particular Supplementary Fig. 13b), although in this case it might require further histological analysis (as in Figure 2h). I leave it to the discretion of the Editor and the Authors, and only if the format allows this modification.

I have no further comments or suggestions. I congratulate the authors for the work performed.

Daniel Muñoz-Espín, PhD

Reviewer #2 (Remarks to the Author):

Wakita et al have now submitted a revised version, adequately addressing all concerns raised by all 3 reviewers.

I have no further comments.

Reviewer #3 (Remarks to the Author):

The authors have adequately addressed all my concerns. I support the publication of the revised manuscript in Nature Communications.

Point-by-point responses to the reviewers' comments

We sincerely thank both the reviewers and the editors for the constructive reviews provided for our manuscript. Our point-by-point responses to the reviewers' comments are detailed below, with the original comments bolded.

Reviewer #1 (Remarks to the Author):

I am pleased with the authors comments and additional experiments as they have addressed the majority of my original concerns.

I would only consider to add Supplementary Fig. 13 to the main text (in particular Supplementary Fig. 13b), although in this case it might require further histological analysis (as in Figure 2h). I leave it to the discretion of the Editor and the Authors, and only if the format allows this modification.

I have no further comments or suggestions. I congratulate the authors for the work performed.

Daniel Muñoz-Espín, PhD

Response:

Since it is difficult to make this modification in its present format, we would like to leave this figure as Supplementary Fig.13b.

Reviewer #2 (Remarks to the Author):

Wakita et al have now submitted a revised version, adequately addressing all concerns raised by all 3 reviewers.

I have no further comments.

Response:

Thank you very much.

Reviewer #3 (Remarks to the Author):

The authors have adequately addressed all my concerns. I support the publication of the revised manuscript in Nature Communications.

Response:

Thank you very much.